



# Microphysical Processes Producing High Ice Water Contents (HIWCs) in Tropical Convective Clouds during the HAIC-HIWC Field Campaign: Evaluation of Simulations Using Bulk Microphysical Schemes

Yongjie Huang[1], Wei Wu[2], Greg M. McFarquhar[1,2], Xuguang Wang[1], Hugh Morrison[3],
Alexander Ryzhkov[2,5], Yachao Hu[2,4], Mengistu Wolde[6], Cuong Nguyen[6], Alfons Schwarzenboeck[7],
Jason Milbrandt[8], Alexei V. Korolev[8], and Ivan Heckman[8]

[1]School of Meteorology, University of Oklahoma, Norman, OK, USA
[2]Cooperative Institute for Mesoscale Meteorological Studies, University of Oklahoma, Norman, OK, USA
[3]Mesoscale and Microscale Meteorology, National Center for Atmospheric Research, Boulder, CO, USA
[4]Department of Atmospheric and Oceanic Sciences, School of Physics, Peking University, Beijing, China
[5]NOAA/OAR National Severe Storms Laboratory, Norman, OK 73072, USA
[6]National Research Council Canada, Ottawa, Canada
[7]Université Clermont Auvergne, CNRS, UMR 6016, Laboratoire de Météor Physique, Clermont-Ferrand, France
[8]Environment and Climate Change Canada, Dorval, Quebec, Canada

**Correspondence:** Wei Wu (weiwu@ou.edu)

**Abstract.** Regions with high ice water content (HIWC), composed of mainly small ice crystals, frequently occur over convective clouds in the tropics. Such regions can have median mass diameters (MMDs) $< 300$ $\mu$m and equivalent radar reflectivities $< 20$ dBZ. To explore formation mechanisms for these HIWCs, high resolution simulations of tropical convective clouds observed on 26 May 2015 during the High Altitude Ice Crystals - High Ice Water Content (HAIC-HIWC) international field

campaign based out of Cayenne, French Guiana, are conducted using the Weather Research and Forecasting (WRF) model with four different bulk microphysics schemes: the WRF single-moment 6-class microphysics scheme (WSM6), the Morrison scheme and the Predicted Particle Properties (P3) scheme with one- and two-ice options. The simulations are evaluated against data from airborne radar and multiple cloud microphysics probes installed on the French Falcon 20 and Canadian National Research Council (NRC) Convair 580 sampling clouds at different heights. WRF simulations with different microphysics

schemes generally reproduce the vertical profiles of temperature, dew-point temperature and winds during this event compared with radiosonde data, and the coverage and evolution of this tropical convective system compared to satellite retrievals. All of the simulations overestimate the intensity and spatial extent of radar reflectivity by over 30% above the melting layer compared to the airborne X-band radar reflectivity data. They also miss the peak of the observed ice number distribution function for $0.1 < D_{max} < 1$ mm. Even though the P3 scheme has a very different approach representing ice, it does not produce greatly

different total condensed water content or better comparison to other observations in this tropical convective system. Mixed-phase microphysical processes at $-10\,°C$ are associated with the overprediction of liquid water content in the simulations with the Morrison and P3 schemes. The ice water content at $-10\,°C$ increases mainly due to the collection of liquid water by ice





particles, which does not increase ice particle number but increases the mass/size of ice particles and contributes to greater simulated radar reflectivity.

## 1 Introduction

High concentrations of small ice particles ingested into jet engines can cause power-loss and damage events (Lawson et al., 1998; Mason et al., 2006). They can also cause air data probe failures (Duviver, 2010). Regions with high ice water content (HIWC), composed of mainly small ice crystals and median mass diameters (MMDs) as low as 300 $\mu$m, frequently occur over
oceanic convective systems (Mason and Grzych, 2011; Ackerman et al., 2015; Leroy et al., 2016b). Such HIWC regions, with relatively low equivalent radar reflectivities ($Z_e$) (often less than 20 dBZ; Mason et al., 2006; Fridlind et al., 2015; Protat et al., 2016; Wolde et al., 2016; Leroy et al., 2017), are hard to detect with pilot radars onboard commercial aircraft and are thus potentially hazardous.

In order to explore the processes responsible for the occurrence of HIWC regions and the associated ice crystal properties
within tropical convection, the High Altitude Ice Crystals - High Ice Water Content (HAIC-HIWC) international field campaigns (Dezitter et al., 2013; Strapp et al., 2016a) and the HIWC RADAR campaign (Ratvasky et al., 2019) were conducted. The first HAIC-HIWC field campaign took place near Darwin, Australia during the monsoon season of 2014, with the second out of Cayenne, French Guiana in May 2015. The HIWC RADAR campaign was conducted out of Florida in August 2015. Data collected during these campaigns have been being analyzed to understand HIWC conditions (Leroy et al., 2015, 2016a,
2017; Protat et al., 2016; Wolde et al., 2016; Korolev et al., 2020), to develop warning products that can identify HIWC regions (Yost et al., 2018; Bedka et al., 2019; Harrah et al., 2019; Haggerty et al., 2020), and to characterize the high altitude HIWC environment to assess a new ice crystal aircraft certification envelope.

In general, total condensed water content (TWC) values in these campaigns reached as high as 4.1 g m$^{-3}$ averaged over 0.93 km (0.5 nautical mile) distance scales, and even up to about 2 g m$^{-3}$ over 185 km distance scales. Average MMDs in
HIWC zones greater than or equal to 1 g m$^{-3}$ increased with temperature, from ~326 $\mu$m at $-50\,^{\circ}$C to ~708 $\mu$m at $-10\,^{\circ}$C (Strapp et al., 2020). Leroy et al. (2015, 2016a, 2017) showed that MMDs decrease with increasing TWC and decreasing temperature, indicating small ice crystals are responsible for HIWC regions at high altitudes for both the Darwin and Cayenne datasets. Wolde et al. (2016) found the relationship between the ice water content (IWC) and radar equivalent reflectivity factor followed a power-law fit with coefficients dependent on temperature. However, they found the pilot X-band weather
radar on the Canadian National Research Council (NRC)'s Convair-580 aircraft did not have adequate sensitivity to detect HIWC regions when calibrated using the NRC X-band research radar. Nguyen et al. (2019) proposed a retrieval method for IWC using the specific differential phase ($K_{dp}$) and differential reflectivity ratio ($Z_{dr}$) data from X-band dual-polarization airborne radar. This method was demonstrated to be superior to the power-law fits between IWC and reflectivity as accounting



for $Z_{dr}$ reduced the dependency of IWC on the variation in the shapes and orientation of ice particles. Ladino et al. (2017)
concluded that secondary ice production (SIP) plays a dominant role in the formation of the observed high concentration ice
crystals with ice nucleating particles making only a minor contribution. Korolev et al. (2020) proposed that a new "freezing-
drop-shattering" mechanism generated small SIP particles above the melting layer at temperatures between 0 and $-15\,^{\circ}$C in
both oceanic tropical mesoscale convective systems (MCSs) and midlatitude frontal clouds. In this SIP mechanism, large liquid
drops are transported through the melting layer to a supercooled environment by convective turbulent updrafts, and then collide
with aged ice, freeze, and shatter.

Several numerical simulation studies on tropical MCSs sampled during the HAIC-HIWC projects have been conducted us-
ing different numerical models and different microphysics schemes. Franklin et al. (2016) showed that the Met Office Unified
Model (UM) with a single-moment microphysics scheme overestimated the radar reflectivity above the freezing level due to
the errors of simulated updraft dynamics and particle sizes, hypothesizing that a double-moment microphysics scheme would
improve model's representation of the observed variability of the ice particle size distribution (PSD). Stanford et al. (2017)'s
WRF simulations of four tropical deep convection events sampled during the HAIC-HIWC Darwin campaign showed that
three microphysics schemes (one bin and two double-moment bulk schemes, Lynn et al., 2005; Thompson et al., 2008; Mor-
rison et al., 2009) produced larger MMDs for TWC $> 1$ g m$^{-3}$ at temperatures between $-10$ and $-40\,^{\circ}$C, and a high bias in
convective radar reflectivity compared to observations. They hypothesized these differences resulted from errors in parameter-
ized hydrometeor PSDs, single ice particle properties (e.g., shape and density) and parameterized microphysical processes. Qu
et al. (2018)'s simulation of a tropical convective system on 16 May 2015 with the Environment and Climate Change Canada's
Global Environmental Multiscale (GEM) model and the Milbrandt–Yau double-moment cloud microphysical scheme (MY2)
(Milbrandt and Yau, 2005) also produced IWC and ice particle number concentration differing from observations during the
HAIC-HIWC Cayenne project, which they hypothesized was due to the poor representation of SIP processes in the micro-
physics scheme.

Other tropical convective clouds have also been observed and simulated. For example, McFarquhar and Heymsfield (1996)
found the numbers of smaller particles ($D < 100\,\mu$m) close to the convection were one order of magnitude higher than the
numbers found further away according to data obtained during the Central Equatorial Pacific Experiment (CEPEX). Mean-
while, McFarquhar and Heymsfield (1997) indicated the shapes of the ice PSDs in the tropics substantially differ from those
in the midlatitudes, especially at temperatures $< -40\,^{\circ}$C. Lohmann et al. (1995) showed that the simulated average IWC by a
coarser resolution ($\sim$125 km $\times$ 125 km) general circulation model (GCM) agreed well with the observed IWC during CEPEX,
especially with respect to the relationship between IWC and temperature, whereas the model underestimated the variability of
simulated IWC within each temperature bin. Chen et al. (1997) indicated the main sources of ice particles are frozen cloud
droplets and interstitial aerosol particles, and the number concentration of ice particles is influenced strongly by the amount of
condensation nuclei in convective inflows according to the simulations and a sensitivity experiment for cases during CEPEX
by using a one-dimensional microphysical model. Ackerman et al. (2015) conducted three 3D cloud-resolving model (CRM)
simulations of MCSs observed on 23 January 2006 during the Tropical Warm Pool International Cloud Experiment (TWP-ICE)
with bulk and bin microphysics schemes, but were not able to produce HIWC regions (IWC $> 2$ g m$^{-3}$ and $Z_e < 30$ dBZ).





Lang et al. (2011) greatly reduced model bias of excessively large reflectivity values (e.g., 40 dBZ) in the middle and upper tro-
posphere in the simulation of a continental convective case observed during the Tropical Rainfall Measuring Mission (TRMM)
Large-Scale Biosphere–Atmosphere Experiment in Amazonia (LBA) through modifying a single-moment bulk microphysics
scheme in the Goddard Cumulus Ensemble model; however, there was much less improvement for an oceanic MCS observed
during the TRMM Kwajalein Experiment (KWAJEX).

As indicated by the review above, the numerical studies on HIWC phenomenon to date have not been able to capture
HIWC phenomenon well. This has been attributed to biases in particle properties, parameterized PSDs, and microphysical
processes. The lack of knowledge about processes generating HIWC regions suggests that further numerical simulations are
needed to explore the microphysical pathways producing HIWCs. Qu et al. (2018) indicated MY2 greatly overestimated the
graupel content and hypothesized that HIWC will be better estimated by the next generation of microphysics schemes (e.g.,
the Predicted Particle Properties (P3) microphysics scheme, Morrison and Milbrandt, 2015). Therefore, this study will be the
first test of the P3 scheme in simulating the HIWC phenomenon and in comparing P3 to other bulk microphysics schemes to
determine whether P3 is better able to predict HIWCs in a high resolution numerical weather prediction (NWP) context. In this
study, a tropical oceanic MCS on 26 May 2015, which was well sampled during the Cayenne field campaign, is simulated at
a high resolution with 1-km horizontal grid spacing. The numerical simulation experiments and their evaluation are described
in this paper. In an upcoming companion paper, attention will be focused on sensitivity experiments varying some parameters
within the microphysics scheme to enhance understanding of processes leading to the formation of small crystals in HIWC
regions.

The next section describes the tropical oceanic MCS sampled on 26 May 2015. Section 3 introduces the collected data
and how they were processed. Section 4 shows the simulated fields and their evaluation against observations. A summary and
conclusions are presented in section 5.

## 2  Case description

The tropical MCS observed on May 26 2015 initiated and developed over the tropical Atlantic Ocean north of Cayenne,
French Guiana. The MCS was not associated with obvious synoptic-scale flow features, such as clearly identifiable highs or
lows. From the soundings at Cayenne, a deep moist absolutely unstable layer (MAUL) existed over the area where the MCS
occurred, and the MAUL was maintained as the MCS developed, consistent with the conceptual model of a MCS proposed by
Bryan and Fritsch (2000). There were mainly easterly (westerly) winds below (above) 350 hPa. The first convection initiated
and developed over the ocean near the coast in the early morning. The convection moved eastward over the course of the day
due to the upper westerly winds. Subsequently, new convective cells continually initiated and developed in a similar location
and gradually moved eastward to merge with old convective cells that were present over the ocean to form a large and long-lived
MCS.

The convective system was sampled by two research aircraft, the French SAFIRE Falcon 20 and Canadian NRC Convair
580, during the HAIC-HIWC field campaign. Figure 1 shows the observed brightness temperature from GOES-13 geostationary





satellite channel 4 (10.8 $\mu$m) at 1045 UTC 26 May 2015, tracks of the two flights (Fig. 1a), as well as the height of the aircraft above mean sea level and air temperature at the flight levels (Fig. 1b). Both aircraft sampled close to the convective core of the storm as shown by the tracks of the aircraft through the lowest cloud-top brightness temperature (Fig. 1a). The SAFIRE Falcon 20 sampled at three height levels, i.e., ~7, ~10 and ~11.5 km, corresponding to temperatures of about −10, −30 and −45 °C, respectively. The NRC Convair 580 sampled mainly at ~7 km and a temperature of around −10 °C (Fig. 1b).

## 3  Data and method

### 3.1  Data

The SAFIRE Falcon 20 was equipped with cloud microphysics instrumentation, including a Cloud Droplet Probe (CDP2), Two Dimensional Stereo Imaging Probe (2D-S), Precipitation Imaging Probe (PIP) and Isokinetic Evaporator Probe (IKP-2, Strapp et al., 2016b). The NRC Convair 580 was equipped with an X-band (9.41 GHz) cloud airborne radar (NAX, Wolde et al., 2016) including three antennae (nadir, zenith and side-looking) and similar cloud microphysics instrumentation. The two optical array probes, 2D-S and PIP, recorded 2D images of ice crystals nominally in the size range of 10–1280 and 100–6400 $\mu$m, respectively. The diode resolutions of 2D-S and PIP are 10 and 100 $\mu$m, respectively. The size distribution data with uncertainty of 10%–100% (Baumgardner et al., 2017) are processed following the general approach described in McFarquhar et al. (2017), with only center-in particles accepted, and corrections for out-of-focus particles (Korolev, 2007), shattered particles (Field and Heymsfield, 2003; Field et al., 2006; Korolev and Field, 2015) and particles partially within the photodiode array applied (Heymsfield and Parrish, 1978). Due to a poorly defined depth of field for small particles (Baumgardner et al., 2012) and the potential of shattered artifacts only n(D) for $D_{max} > 50$ $\mu$m are considered here. Composition PSDs ranging from 0.05 to 12.845 mm merged from the 2D-S and the PIP were derived at a 5-s time resolution with a crossover of 400 $\mu$m between probes (Fontaine et al., 2017; Leroy et al., 2017). The IKP-2 bulk TWC probe was designed specifically for these campaigns to measure the high-speed, high-TWC environment, up to at least 10 g m$^{-3}$ at 200 m s$^{-1}$, with a target accuracy of 20% (Strapp et al., 2016b; Leroy et al., 2017).

Radar reflectivity data from the X-band airborne radar installed on the NRC Convair 580 (Wolde et al., 2016), and TWC measured by the IKP-2, and PSDs measured by the 2D-S and PIP installed on the SAFIRE Falcon 20 and on the NRC Convair 580 are used to statistically evaluate the model simulations. Thermodynamic and wind profiles observed by a radiosonde released at Cayenne and the Cayenne GOES-13 Satellite Cloud Products Data (https://doi.org/10.5065/D6NC5ZX6) are also used.

### 3.2  Model setup

The WRF model Version 4.1.3 (Skamarock et al., 2019) is used to simulate the tropical oceanic MCS event on 26 May 2015. Two one-way nested domains with 3- and 1-km horizontal grid spacing and 51 vertical levels are adopted (Fig. 2). The ERA5 reanalysis data available every 1 hr with 0.25°× 0.25° horizontal grid spacing (https://rda.ucar.edu/datasets/ds633.0) are used





for initial and boundary conditions. The model is run from 0000 to 1800 UTC 26 May 2017 for 18 hr with a spin-up time of the first 6 hr. Physical parameterization schemes include the revised Rapid Radiative Transfer Model (RRTMG) longwave and

shortwave radiation scheme (Iacono et al., 2008), the Yonsei University (YSU) planetary boundary layer (PBL) scheme (Hong et al., 2006), the MM5 similarity surface layer scheme (Beljaars, 1995), and the unified Noah land-surface scheme (Tewari et al., 2004). The cumulus parameterization scheme is not activated in this study.

Four bulk microphysics schemes, namely the WRF single-moment 6-class (WSM6) microphysics scheme (Hong and Lim, 2006), the Morrison double-moment scheme (Morrison et al., 2009) and the Predicted Particle Properties (P3) microphysics

scheme with one- and two-ice options (Morrison and Milbrandt, 2015; Milbrandt and Morrison, 2016) are used for separate simulations. The simulations using the WSM6 and Morrison microphysics schemes are referred to as the WSM6 and MORR runs hereafter, respectively. These microphysics schemes are used because the PSDs of ice species are parameterized differently in these schemes. Both the WSM6 and Morrison schemes predict the mixing ratios of five cloud hydrometeor species, including cloud water, rainwater, cloud ice, snow and graupel[1], while the number mixing ratios for all species except cloud water are also

predicted in the Morrison scheme. The P3 scheme predicts bulk ice properties (e.g., mean particle density) rather than predicting separate species of ice with fixed properties (e.g., cloud ice, snow and graupel). P3 uses one or more "free" ice categories to represent all ice-phase hydrometeors, which can eliminate the unphysical "conversion" processes between different traditional ice categories (Morrison and Milbrandt, 2015; Milbrandt and Morrison, 2016). In this study, the options of one- and two-ice categories in the P3 scheme are used, referred to as P3-1ICE and P3-2ICE hereafter, respectively. Technically P3-1ICE and P3-

2ICE are two configurations of the same scheme, but they have notably different treatments of ice which is the basis on which all the microphysics schemes were chosen, so these are referred to as different schemes. Output data in the model domain d02 with 1-km horizontal grid spacing are analyzed in this study. It should be noted that cloud ice, snow, and graupel in WSM6 and MORR and both two categories of ice in P3-2ICE are treated as ice particles to compare with the observed ice particles, because the observed ice particles are not separated into different categories.

## 3.3  Estimation of X band radar reflectivity

The computations of simulated radar reflectivity are performed using the Rayleigh approximation which is applicable at the X-band given the size of typical ice particles (Ryzhkov et al., 2020). The relations for reflectivity from rain ($Z_r$), graupel ($Z_g$), snow ($Z_s$), and ice ($Z_i$) are derived in detail in Appendix A. The total equivalent radar reflectivity factor ($Z_e$) in units of dBZ can thus be attained using

$$Z_e = 10 \times \log_{10} \left( \frac{Z_r + Z_g + Z_s + Z_i}{1 \text{ mm}^6 \text{ m}^{-3}} \right). \tag{1}$$

[1]The Morrison scheme has an option to represent rimed ice as either hail or graupel which affects the fallspeed and density, and here the option for graupel is chosen.





## 4 Results

### 4.1 Evaluation of simulated sounding, brightness temperature and radar reflectivity

Figure 3 shows a Skew-T plot of the observed and simulated thermodynamic and wind profiles over Cayenne at 1200 UTC 26 May 2015. The observed profiles of air temperature and dew-point temperature show a very moist environment especially

between 800 and 350 hPa. Regardless of the choice of microphysical scheme, the moist environment is well simulated although the layer between 500 and 350 hPa is drier with a maximum dew point depression (T−Td) of ∼5 °C in P3-2ICE and almost zero in the observations. This upper-level drier layer is likely associated with the initial condition from ERA5 reanalysis data (not shown). The MORR scheme is moister between 500 and 350 hPa, which is more consistent with observations as the maximum T−Td is less than 2 °C (Fig. 3). As for the wind profiles, all the simulations predict the observed easterly and

westerly winds in the lower and upper troposphere respectively, though there are some biases around 300 hPa and up to 250 hPa, where simulated and observed winds are in the opposite directions and wind speeds differ by ∼12 m s$^{-1}$.

Figure 4 shows simulated and observed brightness temperatures (BT) at 1045 UTC 26 May 2015. The simulated BT is calculated using the Community Radiative Transfer Model (CRTM, https://www.jcsda.org/jcsda-project-community-radiative-transfer-model) using the assumptions consistent with those in the different microphysics schemes. The storm coverage (BT < 232 K, yellow to

deep red areas) in MORR (∼59% of domain) is larger than that in the observations (∼47% of domain), while WSM6 produces a smaller storm coverage (∼33% of domain) compared to observations. The storm coverages (BT < 232 K) in P3-1ICE (∼51% of domain) and P3-2ICE (∼46% of domain) are closer to that of the observations (Fig. 4). The lower brightness temperature areas representing deep convection (BT < 212 K, red areas in Fig. 4) are larger in MORR (∼25% of domain), P3-1ICE (∼20% of domain), P3-2ICE (∼13% of domain) and smaller in WSM6 (∼5% of domain) than in the observations (∼9% of domain).

To examine the storm evolution, the frequency distributions of simulated and observed BT from 0615 UTC to 1745 UTC 26 May 2015 are displayed in Fig. 5. By 1015 UTC, the dominant frequency (> 4%) of simulated and observed BT is around 280 K, indicating that clear regions dominate the domain in the early stage of MCS. The subdominant frequency (2–4%) around 220 K illustrates the deep convection. This subdominant frequency in all of the simulations is consistent with observations, even though the BT ranges are all slightly different ( i.e., 216–232 K in WSM6, 202–226 K in MORR, 200–228 K in P3-1ICE,

206–228 K in P3-2ICE, and 208–230 K in observations). There are 25% of brightness temperatures < 226 K in WSM6, < 212 K in MORR, < 214 K in P3-1ICE, < 216 K in P3-2ICE, and < 220 K in the observations at 1015 UTC, indicating MORR, P3-1ICE and P3-2ICE overpredict strong convective areas, and WSM6 underpredicts strong convective areas compared to the observations. After 1015 UTC, the frequency around 220 K becomes dominant in the observations and simulations indicating the MCS develops and deep convective areas enlarge, as the maximum frequency exceeds 10% in the observations and in

MORR. The frequency ranges (> 4%) cover 222–246 K in WSM6, 206–234 K in MORR, 208–238 K in P3-1ICE, 210–236 K in P3-2ICE, and 212–232 K in the observations. There are 50% of brightness temperatures < 238 K in WSM6, < 220 K in MORR, < 225 K in P3-1ICE, < 228 K in P3-2ICE and < 227 K in the observation at 1415 UTC. Overall this indicates that the storm coverage in MORR, P3-1ICE and P3-2ICE is more consistent with the observations than that in WSM6. The frequency of brightness temperatures 214–224 K is over 10% after 1230 UTC in MORR, and during 1500–1600 UTC in the





observations. This means deep convective areas are larger in MORR than the observations at an early stage in the system. Overall, all the simulations generally reproduce the storm coverage and evolution of this tropical MCS with the average bias (difference between simulation and observation) in storm coverage (BT < 232 K) of ~−34.3% in WSM6, ~30.0% in MORR, ~12.9% in P3-1ICE, and ~2.3% in P3-2ICE, indicating there is relatively larger bias in WSM6 in which the storm coverage is smaller. WSM6 underestimates the observed deep convection areas (BT < 212 K) by ~55.5% [2] and the overestimates in

MORR, P3-1ICE and P3-2ICE are ~175.4%, ~178.2% and ~76.5%, respectively (Fig. 5).

To obtain a statistical comparison between simulated and observed radar reflectivity, Contoured Frequency by Altitude Diagrams (CFADs) (Yuter and Houze Jr, 1995) are used. Because the observed reflectivity is only available along the flight track, equivalent locations for sampling the reflectivities from the modeled fields must be determined. The sampling method here is based on the flight track and observed and simulated BTs. Because there exists a bias between the simulated and observed BTs,

they are first spatially normalized respectively within the same region as the model domain d02. Using normalized BTs here paired-samples of observations and simulations are found whose locations are about the same distance from the convective cores. Only the vertical profiles of observed reflectivity in which the observed TWC at the flight locations are larger than 0.1 g m$^{-3}$ are selected as observational samples for the statistical comparison. Across an area with the horizontal location of observed reflectivity profiles as the center and 100 km as the range (Fig. 6), the simulated vertical profiles of reflectivity at model

grid points where simulated TWC at the flight level is larger than 0.1 g m$^{-3}$ and the normalized simulated brightness temperature is closest to the normalized observed brightness temperature are selected as simulation samples for statistical comparison. It should be noted that radii from 20 to 200 km in the 20-km intervals were tested, and the results were similar. A radius of 100 km was adopted because the standard deviation between the observed and simulated reflectively was the least when using this radius threshold. The sampling method used here is similar to the one used by Borderies et al. (2018).

In general, the simulations overestimate the X-band radar reflectivity above the melting layer (~4.7 km). Figure 7 shows CFADs and cumulative CFADs of simulated and observed X-band radar reflectivity above 5 km. The CFADs are shown only above 5 km, because the formulae for calculating the simulated radar reflectivity in section 3.3 do not consider the effect of melting, and this study mainly focuses on HIWC regions. From Fig. 7e, 95% of the observed reflectivities are < 30 dBZ above 6 km. The most frequently observed radar reflectivity is around 25 dBZ at heights of 5–7 km and ~15 and ~20 dBZ at a

height of ~8 km (Fig. 7e). The simulated radar reflectivity shows broader distributions with larger values than the observations (95% of the observed reflectivities < 30 dBZ above 6 km), and maxima in radar reflectivity can reach 50 dBZ, especially for WSM6, P3-1ICE and P3-2ICE (Figs. 7a, c and d). There are 95% of the simulated reflectivities < 44 dBZ in WSM6, < 41 dBZ in MORR, < 45 dBZ in P3-1ICE, and < 47 dBZ in P3-2ICE above 6 km. The simulated radar reflectivity in MORR has a narrower distribution with 70% of reflectivities between 34 and 42 dBZ at 5 km (Fig. 7b), which better resembles

the observation with 70% of reflectivities between 24 and 36 dBZ at 5 km (Fig. 7e). The other simulations have broader distributions with 70% of the reflectivities between 30 and 44 dBZ in WSM6 (Fig. 7a), between 17 and 46 dBZ in P3-1ICE (Fig. 7c), and between 25 and 48 dBZ in P3-2ICE (Fig. 7d) at 5 km. The radar reflectivity in all simulations extends above

---

[2]The percentage of underestimate/overestimate/underprediction/overprediction in this paper is the ratio of difference between simulation and observation to observation if not otherwise specified.



14 km, whereas the observed radar reflectivity is mainly below 14 km. Examining the reflectivity and Doppler velocity from zenith-viewing Doppler airborne radar shows a peak-to-peak correlation between them. This suggests there may be stronger updrafts in the simulations associated with the higher extended simulated reflectivity. Overall, all the simulations overestimate the intensity and spatial extent of radar reflectivity. By examining each component of reflectivity, the overestimation of radar reflectivity above the melting layer in WSM6 and MORR mainly results from the overprediction of graupel (not shown), which is similar to the tropical MCS simulations of Lang et al. (2011) and Qu et al. (2018). The P3 scheme, which was expected to yield better estimates of HIWCs, does not reduce the biases in simulated radar reflectivities. It should be noted that these biases may be also related to the aircraft sampling statistics. The NRC Convair 580 operations had to avoid the cloud regions with high reflectivity due to safety regulations, and thus it did not approach high reflectivity regions (red zones on the pilot's radar) and within 30 nautical miles (∼55.56 km). Therefore, sampling statistics is partly biased due to exclusion of cloud regions with high reflectivity.

### 4.2 Cloud microphysical properties

Samples are selected to examine the observed and simulated cloud microphysical properties using the same method as used for sampling radar reflectivity profiles. Compared to the observed ice PSDs, the simulated PSDs have different shapes and variability depending on the microphysics scheme. Figures 8–10 show observed and simulated ice PSDs at the levels of $-45$, $-30$, and $-10$ °C, respectively. These are the three temperature levels at which the in-situ observations were focused. The simulated ice PSDs are the composite PSDs of all the ice-phase hydrometeors predicted in each microphysics scheme.

At the $-45$ °C level (Fig. 8), compared to the observations, WSM6 underestimates the median number distribution function n(D) by ∼50% (∼$3 \times 10^5$ m$^{-3}$ mm$^{-1}$) near the maximum dimension ($D_{max}$) of 0.1 mm (Fig. 8a), while P3-1ICE and P3-2ICE overestimate the median n(D) by ∼282% (∼$17 \times 10^5$ m$^{-3}$ mm$^{-1}$) and ∼199% (∼$12 \times 10^5$ m$^{-3}$ mm$^{-1}$) respectively at this $D_{max}$ (Figs. 8c and d). MORR has similar median n(D) magnitude to the observations with both ∼$6 \times 10^5$ m$^{-3}$ mm$^{-1}$ near the $D_{max}$ of 0.1 mm (Fig. 8b). WSM6, MORR and P3-2ICE underpredict the observed number concentration for 0.1 mm $< D_{max} <$ 12.845 mm (N$_{0.1-12.845mm}$) by ∼17%, ∼50% and ∼16% respectively, while P3-1ICE overpredicts it by ∼5 times (Table 1). Small particles (0.1 mm $< D_{max} <$ 0.3 mm) consistently make dominant contributions to the total number concentration in observations and simulations, while P3-1ICE produces too many small particles (∼86.6%) with the overprediction of total number concentration (Table 1). Compared to the observed PSDs, PSDs in WSM6 and MORR have a smaller spread (Figs. 8a and b), and PSDs in P3-1ICE and P3-2ICE have a larger spread (Figs. 8c and d). Based on the shapes, magnitudes and spreads of PSDs at $-45$ °C, PSDs in MORR are most consistent with the observations among the simulations.

At the $-30$ °C level (Fig. 9), all the shapes of simulated PSDs are similar to those at $-45$ °C, while the magnitudes are smaller by about an order of magnitude (Fig. 8). WSM6 and MORR have similar PSD characteristics in terms of spread and magnitude. The median n(D) in WSM6 and MORR have similar magnitude (∼$10^5$ m$^{-3}$ mm$^{-1}$) at $D_{max}$ of 0.1 mm compared to the observations (Figs. 9a and b). P3-1ICE and P3-2ICE overestimate the median n(D) by ∼6 times (∼$5.7 \times 10^5$ m$^{-3}$ mm$^{-1}$) and ∼264% (∼$2.5 \times 10^5$ m$^{-3}$ mm$^{-1}$) respectively at $D_{max}$ of 0.1 mm (Figs. 9c and d). None of the simulations capture the peak of the observed PSD with the median of ∼$2.6 \times 10^5$ m$^{-3}$ mm$^{-1}$ near $D_{max}$ of 0.3 mm (Fig. 9). There are no obvious





peaks in PSDs for 0.1 mm $< D_{max} <$ 1 mm in WSM6, MORR, and P3-2ICE (Figs. 9a, b, and d). There is a PSD peak with a median n(D) of $\sim 9.6 \times 10^5$ m$^{-3}$ mm$^{-1}$ near $D_{max}$ of 0.17 mm in P3-1ICE (Fig. 9c). Medium particles (0.3 mm $< D_{max} <$ 1 mm) are dominant in the observations and WSM6, while small particles make the main contributions in MORR, P3-1ICE and
P3-2ICE (Table 1).

At the $-10\,^\circ$C level (Fig. 10), all simulations underestimate the median PSD for $D_{max} <$ 1 mm, especially P3-2ICE with an underestimate by $\sim 94\%$ ($\sim 4.0 \times 10^4$ m$^{-3}$ mm$^{-1}$) at $D_{max}$ of 0.1 mm. Large particles contribute 58.6% of N$_{0.1-12.845mm}$ in P3-2ICE, which is very different from the observations (Table 1). Compared to the observations, MORR has almost the same PSD spread and median for $D_{max}$ of 0.1 mm (Fig. 10b). All of the simulations miss the peak of the observed PSD with a
median of $\sim 1.5 \times 10^5$ m$^{-3}$ mm$^{-1}$ near $D_{max}$ of 0.3 mm (Fig. 10).

Overall, the simulated PSDs at the three temperature levels shown in Figs. 8–10 have biases in various degrees with respect to their shapes, magnitudes and spreads compared to observations. The observations are concentrated around the smaller crystal sizes than are the simulations for most of temperature levels expect P3-1ICE at $-45$ and $-30\,^\circ$C. It should be noted that the PSD comparison strongly reflects different assumptions built into the schemes regarding PSD shapes, namely inverse
exponential PSDs are assumed in WSM6 and MORR, whereas a gamma PSD with a shape parameter ($\mu$) that varies with the slope ($\lambda$) following the observations of (Heymsfield, 2003) is assumed for the P3 scheme.

Figure 11 shows statistical distributions of observed and simulated TWC, number concentration for particle sizes of 0.1 mm $< D_{max} <$ 3 mm (N$_{0.1-3mm}$), effective diameter ($D_e$), and vertical velocity using violin plots (or box-percentile plots, Esty and Banfield, 2003) at temperatures of $-10$, $-30$ and $-45\,^\circ$C, respectively. The shaded areas of violin plots represent
the proportion of the samples outlining the kernel probability densities. It should be noted that the number concentrations for 0.2 mm $< D_{max} <$ 3 mm were also examined, and the conclusions are consistent. Thus, only the results using N$_{0.1-3mm}$ are shown here. There are several different definitions of $D_e$ (McFarquhar and Heymsfield, 1998). Given the main purpose here is to compare the particle sizes between observation and simulation for simplicity, the definition of $D_e$ given by,

$$D_e = \frac{\int_{0.1mm}^{12.845mm} \sum_{k=1}^{K} D^3 n_k(D) dD}{\int_{0.1mm}^{12.845mm} \sum_{k=1}^{K} D^2 n_k(D) dD}, \tag{2}$$

is used, where $K$ is the number of ice species. Since $D$ is maximum dimension, only one number distribution function that includes all data is used for the observations, and therefore $K = 1$ for observations.

Generally all the simulations, especially MORR, reproduce the TWC reasonably well at the three temperature levels. In particular at the $-10\,^\circ$C level the 25th and 75th percentiles of TWC in all the simulations cover the same order of magnitude as the observations (Fig. 11a). The differences in N$_{0.1-3mm}$ among the simulations are quite large (Fig. 11b). At the three
temperature levels, WSM6 and especially MORR underestimate the number concentration (Fig. 11b). This is associated with the underpredicted small particles and overpredicted large particles in WSM6 and MORR compared to the observations (Figs. 8–10). Thus, WSM6 and MORR produce larger $D_e$ compared to the observations at the three temperature levels (Fig. 11c). At $-30\,^\circ$C, the median N$_{0.1-3mm}$ values in WSM6 ($\sim 0.4 \times 10^5$ m$^{-3}$) and MORR ($\sim 0.2 \times 10^5$ m$^{-3}$) are underestimated by $\sim 50\%$ and $\sim 75\%$, respectively, consistent with the underestimate of particle number near the peak of the observed PSD
at $D_{max}$ of $\sim 0.3$ mm (Figs. 9a and b). The N$_{0.1-3mm}$ at $-45\,^\circ$C in P3-1ICE is about one order of magnitude larger than



observed (Fig. 11b) mainly due to many more small particles for 0.1 mm $< D_{max} <$ 0.4 mm (Fig. 8c). Similarly, compared to the observations, P3-1ICE overestimates $N_{0.1-3mm}$ at $-30\,°C$, with the median overestimated by $\sim$129% ($\sim10^5$ m$^{-3}$) (Fig. 11b). This is explained by an overestimate of particle number at $D_{max}$ of $\sim$0.1 mm (Fig. 9c). Accordingly, P3-1ICE produces smaller $D_e$ than the observations at $-45$ and $-30\,°C$ with an underestimate of median $D_e$ by $\sim$38% and $\sim$46% respectively

(Fig. 11c). Due to the larger PSD spreads in P3-2ICE at $-45$ and $-30\,°C$ (Figs. 8d and 9d), $N_{0.1-3mm}$ and $D_e$ in P3-2ICE accordingly have a larger spread than the observations (Figs. 11b and c). This occurs even though the spread of TWC between P3-2ICE and observations is similar (Fig. 11a). At $-10\,°C$, values of $N_{0.1-3mm}$ from all of the simulations, in particular P3-2ICE, are about one order of magnitude smaller than observed (Fig. 11b), implying larger mean particle size than observed (Fig. 11c). This is mainly attributed to the underestimate of small particle number for $D_{max} < 1$ mm, especially near the peak

of the observed PSD at $D_{max}$ of $\sim$0.3 mm, and overestimate of large particles in all of the simulations (Fig. 10). The simulated vertical velocity is in general stronger than in the observations, especially at $-45$ and $-10\,°C$ (Fig. 11c), corresponding to the higher extent of simulated radar reflectivity (Fig. 7).

### 4.3 Cloud microphysical processes

As discussed above, all the four microphysics schemes underpredict the number concentration by about one order of magnitude

at $-10\,°C$ compared to the observations (Fig. 11b), although they predict similar TWC to the observed TWC (Fig. 11a). In this section, the processes producing HIWCs are determined. WSM6 is a single-moment scheme in which the number concentration of ice particles is not predicted directly. Through examining the number concentration derived diagnostically from the water content of each hydrometeor species in WSM6, it is found that their distributions are similar to those in MORR, especially the distributions of ice particles (not shown). Thus, only the double-moment MORR and P3 schemes are examined in detail

here. Regions with IWC $> 1$ g m$^{-3}$ are defined as HIWC regions in this study. There are limited observed and simulated HIWC samples at temperatures of $-45$ and $-30\,°C$ (Fig. 11a). Through examining the microphysical processes of ice particle production using the samples with TWC $> 0.1$ g m$^{-3}$ at $-45$ and $-30\,°C$, it is found that the main microphysical processes at $-45$ and $-30\,°C$ are the same as those within profiles containing HIWC regions at $-10\,°C$. Hence, the profiles of water content, number concentration, and microphysical processes with HIWC regions at $-10\,°C$ are used as examples to discuss

here. The lack of small ice crystals in HIWC regions at this temperature would extend to a lack of small ice crystals at higher altitudes. The subsamples whose observed and corresponding simulated TWCs at $-10\,°C$ are larger than 1 g m$^{-3}$ are selected from the total samples (1778) at $-10\,°C$ (Fig. 10) to conduct a composite analysis. There are 509, 488 and 427 paired samples selected for MORR, P3-1ICE and P3-2ICE, respectively (Table 2).

From Table 2, for the observation–MORR paired HIWC samples the average observed and simulated TWCs at $-10\,°C$ are

similar, $\sim$1.6 and $\sim$1.8 g m$^{-3}$, respectively, while the simulated $N_{0.1-3mm}$ ($\sim 1.78\times10^4$ m$^{-3}$) is about one order of magnitude less than the observed $N_{0.1-3mm}$ ($\sim 1.72\times10^5$ m$^{-3}$). The average simulated air vertical velocity for the HIWC points ($\sim$0.32 m s$^{-1}$) is about twice as large as the observations ($\sim$0.16 m s$^{-1}$). For the observation–P3-1ICE paired samples, the average observed and simulated TWCs at $-10\,°C$ are $\sim$1.7 and $\sim$1.8 g m$^{-3}$ respectively, while the simulated $N_{0.1-3mm}$ ($\sim 1.25\times10^4$ m$^{-3}$) is underpredicted compared to the observed $N_{0.1-3mm}$ ($\sim 1.98\times10^5$ m$^{-3}$) by $\sim$94%. The simulated air vertical velocity





($\sim$1.12 m s$^{-1}$) is 3.67 times greater than observed ($\sim$0.24 m s$^{-1}$). Similarly, for the observation–P3-2ICE paired samples, the averaged observed and simulated TWCs at $-10\,°$C are $\sim$1.7 and $\sim$1.9 g m$^{-3}$ respectively, while the simulated N$_{0.1-3\text{mm}}$ ($\sim$4.63 $\times 10^3$ m$^{-3}$) is about two orders of magnitude less than the observed N$_{0.1-3\text{mm}}$ ($\sim$2.02 $\times 10^5$ m$^{-3}$), and the simulated air vertical velocity ($\sim$2.05 m s$^{-1}$) is about 15 times larger than the observation ($\sim$0.13 m s$^{-1}$). Therefore, both MORR and P3, in particular P3-2ICE, substantially underpredict the ice particle number for 0.1 mm $< D_{max} < 3$ mm and overpredict the

vertical motion in the HIWC regions, which results in stronger and higher-extended simulated radar reflectivity (Fig. 7).

From the vertical profiles of water content for each hydrometeor class in MORR (Fig. 12a), graupel is dominant above the 0 °C layer up to $\sim$8-km height, with snow being second most important. Snow is the dominant hydrometeor category above 8 km in terms of mass content. The aforementioned underestimate of ice particle number concentration at $-10\,°$C is associated with the total number concentration ($< 4 \times 10^4$ m$^{-3}$) of cloud ice, snow and graupel (Fig. 12b). According to Eq. (A17), under the

condition of similar simulated and observed TWCs, an underestimate of ice particle number concentration, especially graupel, leads to large reflectivities. Moreover, not only is the vertical motion overpredicted in the HIWC regions as mentioned above, but the height with air vertical velocity $> 0$ m s$^{-1}$ in MORR can be up to 15 km with a maximum of $\sim$1 m s$^{-1}$ at 13 km (Figs. 12a and b). These characteristics are consistent with the stronger intensity and higher distribution of simulated radar reflectivity shown in Fig. 7b.

The total ice number concentration in P3-1ICE is $\sim$10$^4$ m$^{-3}$ at $-10\,°$C, and the maximum is $\sim$4 $\times 10^5$ m$^{-3}$ at $-45\,°$C (Fig. 12d). The air vertical velocity $> 0$ m s$^{-1}$ in P3-1ICE extends from heights of 1.5 to 15.5 km with a maximum of $\sim$1.4 m s$^{-1}$ at 12.5 km (Figs. 12c and d), which is stronger than MORR. Compared to P3-1ICE, there is one more "free" ice category considered in P3-2ICE. The difference in mean mass-weighted diameters between the two ice categories is over 500 $\mu$m. The vertical motion in P3-2ICE is stronger with a maximum of $\sim$2.1 m s$^{-1}$ at 6 km (Fig. 12e). Although the IWCs are similar between P3-1ICE and P3-2ICE, the total number concentration of the two ice categories in P3-2ICE is less than

$5 \times 10^3$ m$^{-3}$ at $-10\,°$C, which is less than that in P3-1ICE (Figs. 12d and f). The average liquid water contents (LWCs) in MORR, P3-1ICE and P3-2ICE at $-10\,°$C are $\sim$0.029, $\sim$0.094 and $\sim$0.187 g m$^{-3}$, respectively, while the observed LWC from Cloud Droplet Probe (CDP) is less than 0.008 g m$^{-3}$. Thus, all of the simulations, especially P3-2ICE, overpredict LWC at $-10\,°$C and air vertical velocity above the 0 °C layer. The distributions of vertical velocity (Fig. 11d) and Doppler velocity

from zenith-viewing Doppler airborne radar (not shown) confirm that model produces stronger updrafts than observations. It should be noted that there may exist potential biases of aircraft sampling statistics of cloud microphysical parameters, because both the NRC Convair 580 and SAFIRE Falcon 20 avoided cloud regions with strong updrafts where presence of liquid phase is expected.

From vertical profiles of microphysical conversion rates in MORR (Fig. 13), the main source terms of cloud ice content

are ice nucleation at $-45\,°$C and vapor deposition at $-30$ and $-10\,°$C, and the main sink terms are collection by snow and autoconversion to snow. The net conversion rate of cloud ice (Qi_TEND, sum of all microphysical conversion rates including sedimentation) at $-10\,°$C is negative (Fig. 13a). The net number concentration tendency of cloud ice (Ni_TEND) at $-10\,°$C in MORR is $\sim$$-20$ m$^{-3}$ s$^{-1}$, mainly due to the accretion of cloud ice, autoconversion to snow and sublimation (Fig. 13d). The IWC at $-10\,°$C mainly consists of snow and graupel (Fig. 12a), and their main source terms are vapor deposition and



collection of cloud water (Figs. 13b and c). The net snow conversion rate (Qs_TEND) is $\sim 0.1 \times 10^{-3}$ g m$^{-3}$ s$^{-1}$ at $-10\,°$C (Fig. 13b), while there is a negative net conversion rate of graupel (Qg_TEND) ($\sim -0.6 \times 10^{-3}$ g m$^{-3}$ s$^{-1}$) mainly due to its large sedimentation tendency at $-10\,°$C with a rate of $\sim -0.8 \times 10^{-3}$ g m$^{-3}$ s$^{-1}$ (Fig. 13c). Autoconversion of cloud ice to snow ($\sim 5$ m$^{-3}$ s$^{-1}$) and the sedimentation term ($\sim 1$ m$^{-3}$ s$^{-1}$) increase the snow particle number, however, they are offset by the self-collection of snow (aggregation), collection of cloud water by snow to form graupel, and sublimation (Fig. 13e).

The collection of cloud water by snow to form graupel is the main production term of graupel particle number, while it is offset by the sedimentation and sublimation terms. Finally, the net number concentration tendencies of both snow and graupel (Ns_TEND and Ng_TEND) are near 0 m$^{-3}$ s$^{-1}$ at $-10\,°$C (Figs. 13e and f). Therefore, the collection of cloud water by graupel is the key source term of total IWC at $-10\,°$C in MORR, which increases the mean mass/size of graupel and does not directly influence its number. This is associated with the strong simulated reflectivity above the melting layer.

From vertical profiles of microphysical conversion rates in P3-1ICE (Fig. 14a), the main production terms of ice content are vapor deposition at $-45$ and $-30\,°$C, collection of cloud water by ice, vapor deposition and collection of rain water by ice at $-10\,°$C. As for the first ice category in P3-2ICE, in addition to the same main production terms as in P3-1ICE, there is another source term merging from the second ice category due to similar mean mass-weighted diameters between the two ice categories (Fig. 14c). The net tendency of the two ice categories in P3-2ICE (i.e., Qi_TEND + Qi2_TEND) is $\sim 1.05 \times 10^{-3}$ g m$^{-3}$ s$^{-1}$

at $-10\,°$C, which is much larger than that in P3-1ICE ($\sim 0.02 \times 10^{-3}$ g m$^{-3}$). It is mainly due to the stronger collection of cloud water and rain water by ice in P3-2ICE, which may be associated with the greater cloud water and rain water content at $-10\,°$C in P3-2ICE than P3-1ICE (Figs. 12c and e). The aforementioned collection of cloud water and rain water by ice does not increase the ice particle number. Although the deposition nucleation can increase the ice particle number in P3-1ICE and P3-2ICE, it is small (less than 0.5 m$^{-3}$ s$^{-1}$) at $-10\,°$C. Merging ice categories does not increase the total ice particle number

in P3-2ICE. The sedimentation of ice number is $\sim 3.3$ and $\sim 3.4$ m$^{-3}$ s$^{-1}$ at $-10\,°$C in P3-1ICE and P3-2ICE respectively, which dominates the net ice number concentration tendencies at $-10\,°$C in both P3-1ICE ($\sim 2.8$ m$^{-3}$ s$^{-1}$) and P3-2ICE ($\sim 3.1$ m$^{-3}$ s$^{-1}$). The much lower number concentration in P3-2ICE than P3-1ICE (Table 1) is likely due to aggregation associated with collection between the two ice categories in P3-2ICE (Fig. 14).

To summarize briefly, due to the overprediction of LWC in MORR, P3-1ICE and P3-2ICE above the melting layer, there exist

obvious mixed-phase processes at $-10\,°$C. The IWC at $-10\,°$C increases mainly due to the collection of liquid water by ice particles, which does not increase ice particle number but increases the size of ice particles. The lower ice particle numbers in the simulations could also be associated with excessive aggregation and/or missing SIPs,such as collision-induced breakup and "freezing-drop-shattering" proposed by Korolev et al. (2020). The large ice particles and lower ice particle numbers contribute to strong simulated radar reflectivity. Introduction of parameterizations for the missing SIPs may be able to overcome some of

these model limitations, as will be examined in the future.





## 5 Summary and conclusions

A tropical oceanic convective system observed on 26 May 2015 during the HAIC-HIWC international field campaign based out of Cayenne, French Guiana was simulated using the WRF model. Observation data from radiosondes, GOES-13 geostationary satellite, airborne radar, and cloud microphysics instrumentation were used to assess the simulated convective system in terms

of the thermodynamic and dynamic environment, storm coverage, evolution and structure, and microphysical properties. The major results are summarized as follows:

(1) By comparing simulated and observed soundings, all of simulations using different microphysics schemes replicate temperature with average bias within 1.6%, dew-point temparture with average bias within 6%, wind speed with average bias within 14% and wind direction with average bias within 36°, with the MORR scheme giving closest agreement with

observations.

(2) WRF basically reproduces the coverage and evolution of this tropical MCS based on a comparison between simulated and observed brightness temperature with the average bias in storm coverage (brightness temperature < 232 K) by $\sim -34.3\%$ in WSM6, $\sim 30.0\%$ in MORR, $\sim 12.9\%$ in P3-1ICE, and $\sim 2.3\%$ in P3-2ICE. Thus, WSM6 underestimates the storm coverage, and P3-2ICE produces the closest storm coverage to the observation.

(3) In general, all of the simulations overestimate the intensity and spatial extent of radar reflectivity above the melting layer compared to the observed X-band radar reflectivity. There are 95% of the simulated reflectivities < 44 dBZ in WSM6, < 41 dBZ in MORR, < 45 dBZ in P3-1ICE, and < 47 dBZ in P3-2ICE above 6 km, while 95% of the observed reflectivities are < 30 dBZ above 6 km. The radar reflectivity > 0 dBZ in all simulations extends above 14 km, whereas the observed radar reflectivity > 0 dBZ is mainly below 14 km.

(4) Different microphysics schemes have different shapes, magnitudes and spreads in the simulation of ice PSDs at the different temperature levels. All of the simulations miss the peak of the observed ice number distribution function for $0.1 < D_{max} < 1$ mm.

(5) Both the WSM6 and MORR schemes underestimate the number concentration of ice particles at the temperature levels of $-45$, $-30$ and especially $-10$ °C with a maximum bias up to one order of magnitude, though the simulated total water

contents are similar to the observations. This indicates that WSM6 and MORR simulate fewer small particles and more large particles compared to the observations. P3-1ICE overestimates the number concentration with a maximum bias up to one order of magnitude and P3-2ICE generates larger spread of number concentrations covering about two orders of magnitude at temperatures of $-45$ and $-30$ °C. Both schemes, and especially P3-2ICE, underestimate the number concentration by about one order of magnitude at $-10$ °C. This indicates the P3 scheme produces more large particles at this level.

(6) Mixed-phase processes play an important role at $-10$ °C due to the overprediction of LWC in MORR, P3-1ICE and P3-2ICE above the melting layer. Stronger simulated radar reflectivity in MORR above the melting layer results from large graupel particles associated with greater graupel water content and fewer graupel particles compared with in-situ observations. Rapid growth of graupel mass mainly through collecting cloud water but with limited increase in graupel number mainly by conversion of snow to graupel through collection of cloud water above the melting layer leads to large mean graupel sizes in





MORR. Similarly, in P3-1ICE and P3-2ICE the IWC at −10 °C increases mainly due to the collection of cloud water and rain water while the net ice number concentration tendencies are near 0, which generates large mean ice particle sizes. The large ice particles generate strong radar reflectivity, partially explaining the bias of simulated radar reflectivity with P3-1ICE and P3-2ICE.

It should be noted that simulations of deep convection at different model resolutions can be much different. To examine the

sensitivity of model resolution, CFADs and cumulative CFADs of radar reflectivity are also calculated using the simulation data from the 3-km domain (Fig. 15). Although there are some differences in specific values of reflectivity, the intensity and distribution of reflectivity from the 3-km simulations (Fig. 15) are basically consistent with those from the 1-km simulations (Fig. 7). Although this does not prove that the conclusions with respect to the differences in behavior among the microphysics schemes will be the same at all resolutions, it does at least indicate that the results in this study have some generality. It provides

something useful about the microphysics schemes to numerical forecast guidance for HIWCs in current high-resolution NWP models, which are now routinely run at $O(3$ km$)$ and now more and more often at $O(1$ km$)$. However, a caveat is that 1 km is not cloud-resolving $O(100$ m$)$, thus horizontal entrainment is still not being resolved at this resolution, which affects the amount of liquid available for riming growth in updrafts. Therefore, there still exist uncertainties in the 1-km simulations.

In conclusion, the Morrison and P3 microphysics schemes generally outperform the WSM6 scheme in simulating this trop-

ical oceanic MCS as evident from examining the simulated soundings, brightness temperature, radar reflectivity, ice particle size distributions, total water content and number concentration. However, the Morrison scheme underestimates the number concentration at different temperature levels compared to the observations. This indicates that large ice particles, especially graupel, are overpredicted in this scheme, which is similar to Qu et al. (2018)'s simulation of a different tropical MCS using a different model and microphysics scheme. Even though the P3 scheme has a much different approach for representing ice,

it does not produce greatly different TWC or better comparison to the observations using either one- or two-ice categories. This suggests that other aspects need to be considered, such as microphysical process rate formulations or parameters. To enhance understanding of processes leading to the formation of small crystals in HIWC regions, sensitivity experiments varying parameters within the P3 microphysics scheme (e.g., mass-dimensional relations, size distribution parameters, microphysical conversion rates or representation of different processes like secondary ice production) will be examined in a future paper.

*Code and data availability.* The WRF code is available at https://github.com/wrf-model/WRF. Observation data are available at https://data. eol.ucar.edu/master_lists/generated/haic-hiwc_2015. ERA5 reanalysis data are available at https://rda.ucar.edu/datasets/ds633.0.

**Appendix A: Formulae for X band radar reflectivity factor**

Most cloud models with bulk microphysics parameterizations predict either mass content (one-moment schemes) or mass content and number concentration (two-moment schemes) for a number of hydrometeor categories. In one-moment schemes,

number concentration can be obtained diagnostically. They also commonly assume inverse exponential or gamma size distri-





butions of hydrometeors with respect to particle maximum dimension. This allows the use of simple analytical formulae for converting mass content and number concentration to radar reflectivity factor if the scatterers are small compared to the radar wavelength so that the Rayleigh approximation can be used. A gamma size distribution is represented by

$$N(D) = N_0 D^\mu e^{-\Lambda D}, \tag{A1}$$

where $N_0$, $\mu$ and $\lambda$ are the intercept, shape and slope factors respectively. The total number concentration $N_t$ is thus given by

$$N_t = \int_0^\infty N(D)dD = \frac{N_0 \Gamma(\mu+1)}{\Lambda^{\mu+1}}. \tag{A2}$$

**A1   Calculation of radar reflectivity factor for Rain**

For rain, the radar reflectivity factor $Z$ in the Rayleigh approximation is given by the well known formula

$$Z = \int_0^\infty D_{eq}^6 N(D_{eq})dD_{eq}, \tag{A3}$$

where $D_{eq}$ is the equivolume diameter of raindrop, and $D_{eq} = D$ for liquid water species in bulk schemes. The liquid water content $LWC$ is defined as

$$LWC = \frac{\pi}{6}\rho_w \int_0^\infty D^3 N(D)dD, \tag{A4}$$

where $\rho_w$ is the density of water. Taking into account Eq. (A2), integration of Eq. (A3) and Eq. (A4) yields

$$Z = \frac{N_t \Gamma(\mu+7)}{\Lambda^6 \Gamma(\mu+1)}, \tag{A5}$$

and

$$LWC = \frac{\pi \rho_w N_t \Gamma(\mu+4)}{6\Lambda^3 \Gamma(\mu+1)}, \tag{A6}$$

which results in the following formula for estimating $Z$ from $LWC$ and $N_t$ with an inverse exponential size distribution assumption ($\mu = 0$) in WSM6, MORR and P3:

$$Z = \frac{720 LWC^2}{\pi^2 \rho_w^2 N_t}. \tag{A7}$$

**A2   Calculation of radar reflectivity factor for ice particles**

The radar reflectivity factor for ice is defined as

$$Z = \frac{|K_i|^2}{|K_w|^2 \rho_i^2} \int_0^\infty \rho_s^2(D_{eq}) D_{eq}^6 N(D_{eq})dD_{eq}, \tag{A8}$$





(Ryzhkov and Zrnić, 2019, Eq. (5.14)), where $\rho_i$ is the density of solid ice sphere, assumed here to be 0.917 g cm$^{-3}$, and

$$K_w = \frac{\varepsilon_w - 1}{\varepsilon_w + 2}, \text{and } K_i = \frac{\varepsilon_i - 1}{\varepsilon_i + 2},$$ (A9)

where $\varepsilon_w$ and $\varepsilon_i$ are the dielectric constants of water and solid ice, and $|K_w|^2 = 0.930$ and $|K_i|^2 = 0.176$ in this study. Taking into account that the mass of ice particle $m_s$ can be expressed as

$$m_s(D_{eq}) = \frac{\pi}{6} \rho_s(D_{eq}) D_{eq}^3,$$ (A10)

Eq. (A8) can be written in a different form often used in cloud models (e.g., Hogan et al., 2006):

$$Z = \frac{|K_i|^2}{|K_w|^2} \left( \frac{6}{\pi \rho_i} \right)^2 \int_0^\infty m_s^2(D_{eq}) N(D_{eq}) dD_{eq},$$ (A11)

where $D_{eq}$ is the equivalent volume diameter of an ice particle (Ryzhkov and Zrnić, 2019).

The $m_s$–$D_{max}$ relations are commonly used (e.g., Locatelli and Hobbs, 1974; Mitchell, 1996; Finlon et al., 2019; Ding et al., 2020) where $D_{max}$ is the maximal dimension of ice particles. These relations are often represented as power-law dependencies

$$m_s(D_{max}) = a D_{max}^b.$$ (A12)

Then Eq. (A11) can be rewritten as

$$Z = \frac{|K_i|^2}{|K_w|^2} \left( \frac{6}{\pi \rho_i} \right)^2 \int_0^\infty a^2 D_{max}^{2b} N(D_{eq}) dD_{eq} = \frac{|K_i|^2}{|K_w|^2} \left( \frac{6}{\pi \rho_i} \right)^2 \int_0^\infty a_0^2 D_{eq}^{2b} N(D_{eq}) dD_{eq},$$ (A13)

where $a_0 = a\eta^b$. In Eq. (A13), the difference between maximal dimension $D_{max}$ and equivolume diameter $D_{eq}$ is taken into account with a scaling factor $\eta = D_{max}/D_{eq}$ assuming a constant aspect ratio of ice particles across the size spectrum.

### A2.1 Constant $m$–$D$ relation across the size spectrum

In WSM6 and MORR, $a$ and $b$ in the $m_s$–$D_{max}$ relations are assumed to be constant across the size spectrum for snow and graupel particles, implying densities of snow and graupel are constant. This leads to the following expressions for $Z$ and ice water content $IWC$:

$$Z = \frac{|K_i|^2}{|K_w|^2} \left( \frac{6}{\pi \rho_i} \right)^2 \frac{\Gamma(2b + \mu + 1) N_t}{\Gamma(\mu + 1)} \frac{a_0^2}{\Lambda^{2b}}$$ (A14)

and

$$IWC = \int_0^\infty a_0 D_{eq}^b N(D_{eq}) dD_{eq} = \frac{\Gamma(b + \mu + 1) N_t}{\Gamma(\mu + 1)} \frac{a_0}{\Lambda^b},$$ (A15)

so that

$$Z = \frac{|K_i|^2}{|K_w|^2} \left( \frac{6}{\pi \rho_i} \right)^2 \frac{\Gamma(\mu + 1) \Gamma(2b + \mu + 1)}{[\Gamma(b + \mu + 1)]^2} \frac{IWC^2}{N_t}.$$ (A16)



It is important that Eq. (A16) is not sensitive to the variability of the prefactor $a$ in the $m_s(D_{max})$ power-law relation (A12) and is minimally affected by the variability of the exponent $b$ in Eq. (A12). In WSM6 and MORR, $b = 3$ and $\mu = 0$, thus Eq.

(A16) can be simplified as

$$Z = \frac{720}{\pi^2 \rho_i^2} \frac{|K_i|^2}{|K_w|^2} \frac{IWC^2}{N_t}. \tag{A17}$$

**A2.2 Variable $m$–$D$ relation across the size spectrum**

P3 represents each ice category much differently than WSM6 and MORR. In P3, $a$ and $b$ in the $m_s$–$D_{max}$ relations [Eq. (A12)] are variable across the size spectrum (Morrison and Milbrandt, 2015). Therefore, radar reflectivity factor is calculated

numerically by using Eq. (A13).

*Author contributions.* YH, WW, GMM designed the study. YH did the calculations, with support from WW, GMM, XW and HM. AR developed the method for calculating X-band radar reflectivity. MW, CN, AS, AVK, and IH processed the original observational datasets. YH wrote the original draft with contributions from all coauthors, and all coauthors contributed to review and editing.

*Competing interests.* The authors declare that they have no conflict of interest.

*Acknowledgements.* This work was supported by the National Science Foundation (Award Numbers: 1213311 and 1842094). Observational data are provided by NCAR/EOL under the sponsorship of the National Science Foundation (https://data.eol.ucar.edu/). The authors are grateful to NCAR's Data Support Section for providing ERA5 reanalysis data (https://rda.ucar.edu/datasets/ds633.0). The authors acknowledge high-performance computing support from Cheyenne (https://doi.org/10.5065/D6RX99HX) provided by NCAR's Computational and Information Systems Laboratory, sponsored by the National Science Foundation. NCAR is sponsored by the National Science Foundation.

Some of the computing for this project was performed at the University of Oklahoma (OU) Supercomputing Center for Education and Research (OSCER). The discussions of radar forward simulators with Jiaxi Hu and Djordje Mirkovic, and the discussions of HIWC conditions and IKP with Walter Strapp are greatly appreciated. Major North American funding for flight campaigns was provided by the FAA William Hughes Technical Center and Aviation Weather Research Program, the NASA Aeronautics Research Mission Directorate Aviation Safety Program, the Boeing Co., Environment and Climate Change Canada, the NRC of Canada, and Transport Canada. Major European campaign

and research funding was provided from (i) the European Commission Seventh Framework Program in research, technological development and demonstration under grant agreement n°ACP2-GA-2012-314314, (ii) the European Aviation Safety Agency (EASA) Research Program under service contract n° EASA.2013.FC27. Further funding was provided by the Ice Crystal Consortium.





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



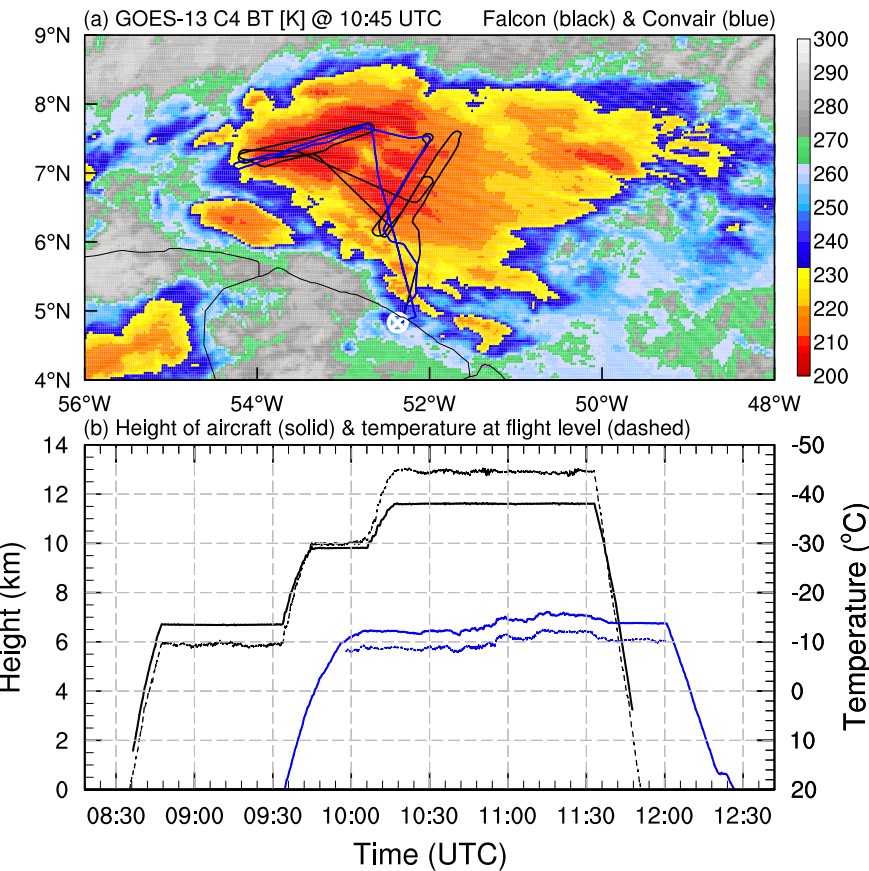

**Figure 1.** (a) Observed brightness temperature (K, shaded) from GOES-13 geostationary satellite channel 4 (10.8 $\mu$m) at 1045 UTC 26 May 2015 and flight tracks of SAFIRE Falcon 20 (black thick line) and NRC Convair 580 (blue thick line). (b) Height (km, thick solid curves) of aircraft above mean sea level and air temperature (°C, thin dashed curves) at the flight level for SAFIRE Falcon 20 (black) and NRC Convair 580 (blue). The white mark "⊗" in (a) represents the location of the release of a radiosonde at Cayenne.



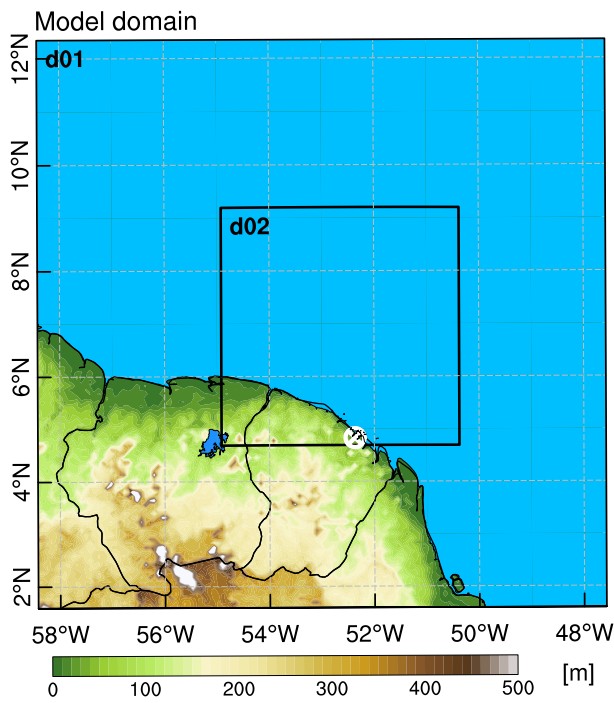

**Figure 2.** The model domain configuration (color shaded fields represent terrain elevation, in m). The horizontal grid spacings of d01 and d02 are 3 and 1 km, respectively. The white mark "⊗" represents the location of the release of a radiosonde at Cayenne.

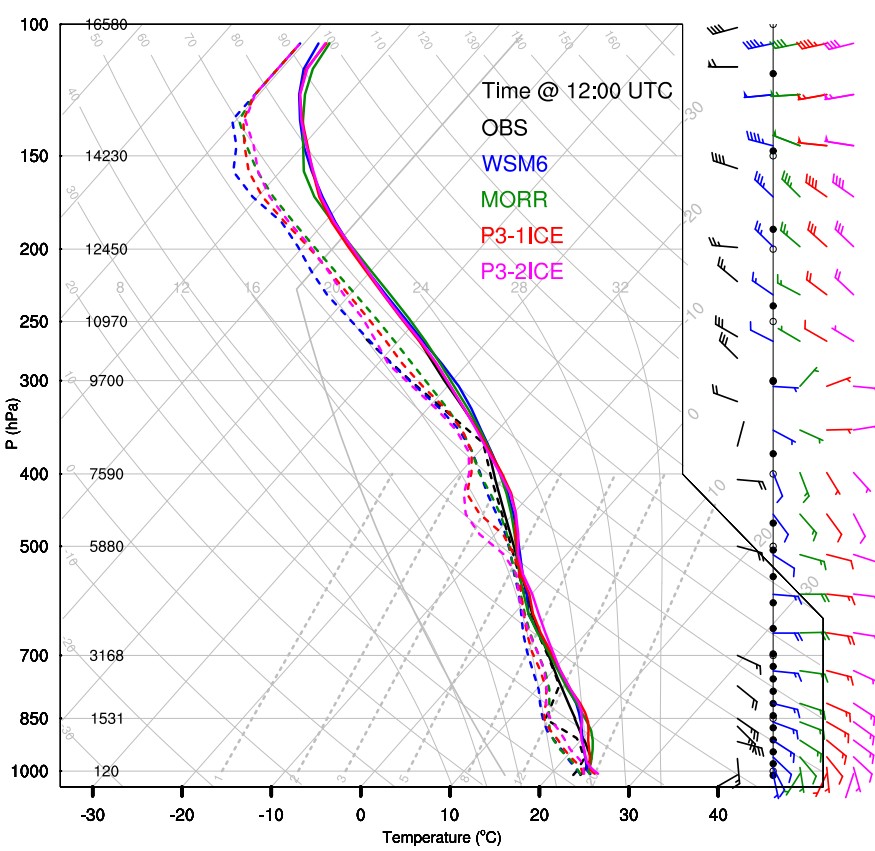

**Figure 3.** Skew-T plot of observed and simulated Cayenne radiosonde data at 1200 UTC 26 May 2015. One full wind barb represents 10 knot ($\sim$5.14 m s$^{-1}$). The location where Cayenne radiosonde was released is shown in Fig. 2.

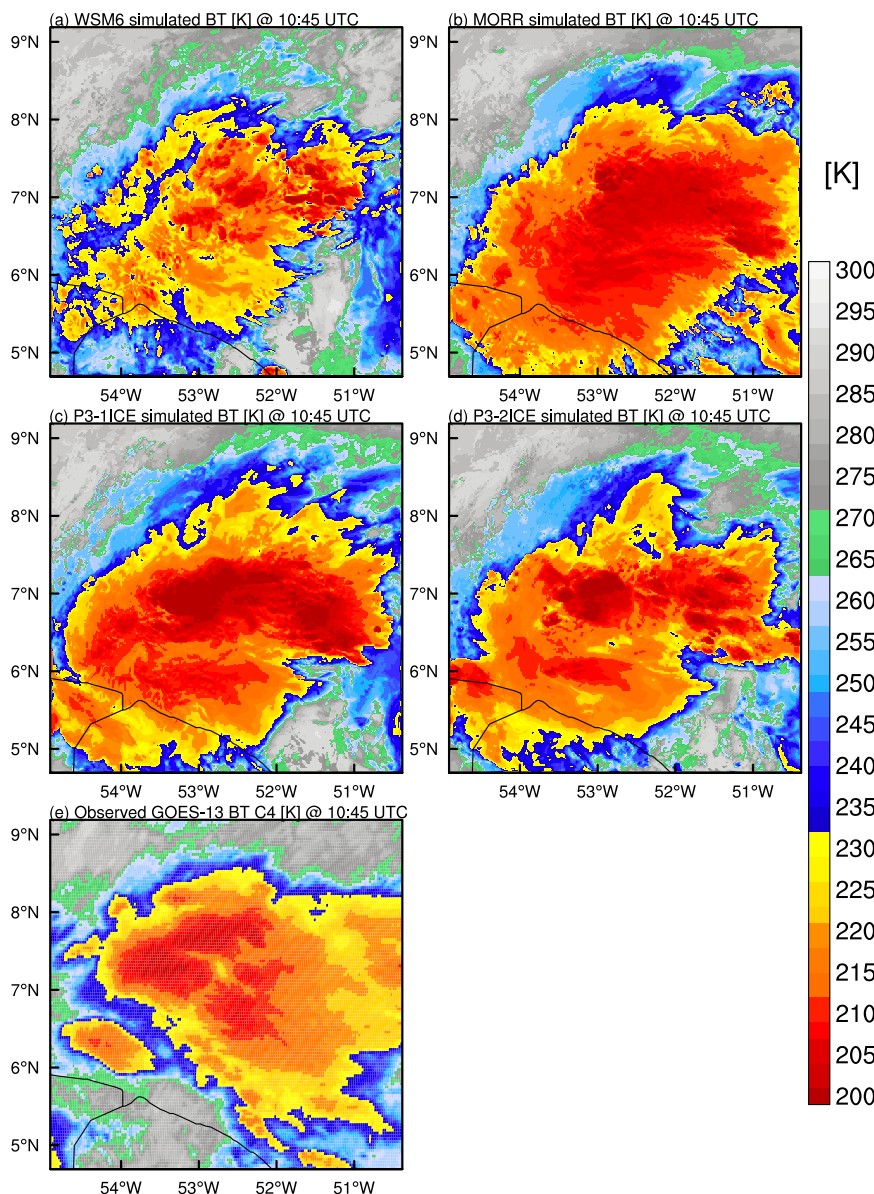

**Figure 4.** Simulated and observed brightness temperature (BT, K, shaded) from GOES-13 channel 4 (10.8 $\mu$m) at 1045 UTC 26 May 2015.





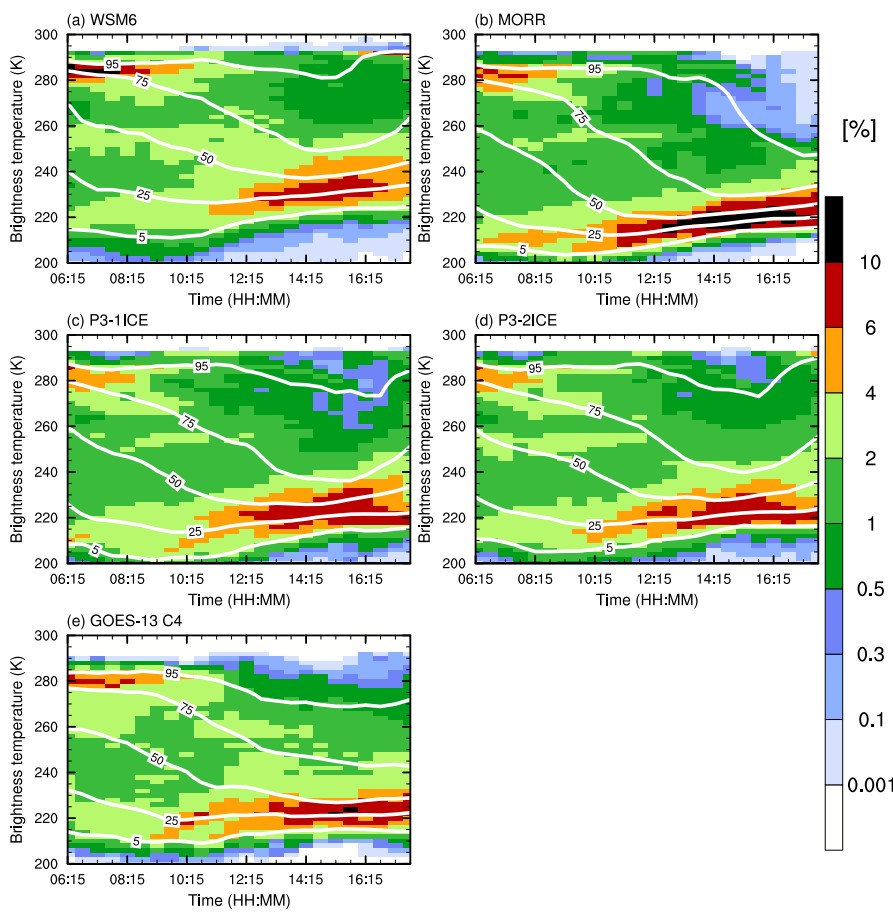

**Figure 5.** Frequency (%, shaded) of simulated and observed BT from GOES-13 channel 4 from 0615 to 1745 UTC 26 May 2015. Contours represent cumulative frequencies 5%, 25%, 50%, 75% and 95%, respectively.





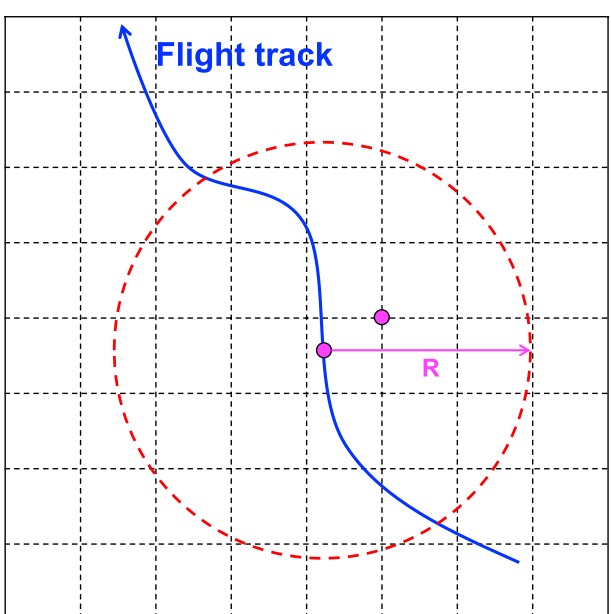

**Figure 6.** Diagram of sampling method for radar reflectivity profiles. R = 100 km here. The two magenta points at the flight track and model grid point represent the horizontal locations of observed and simulated radar reflectivity profiles. See text for details.

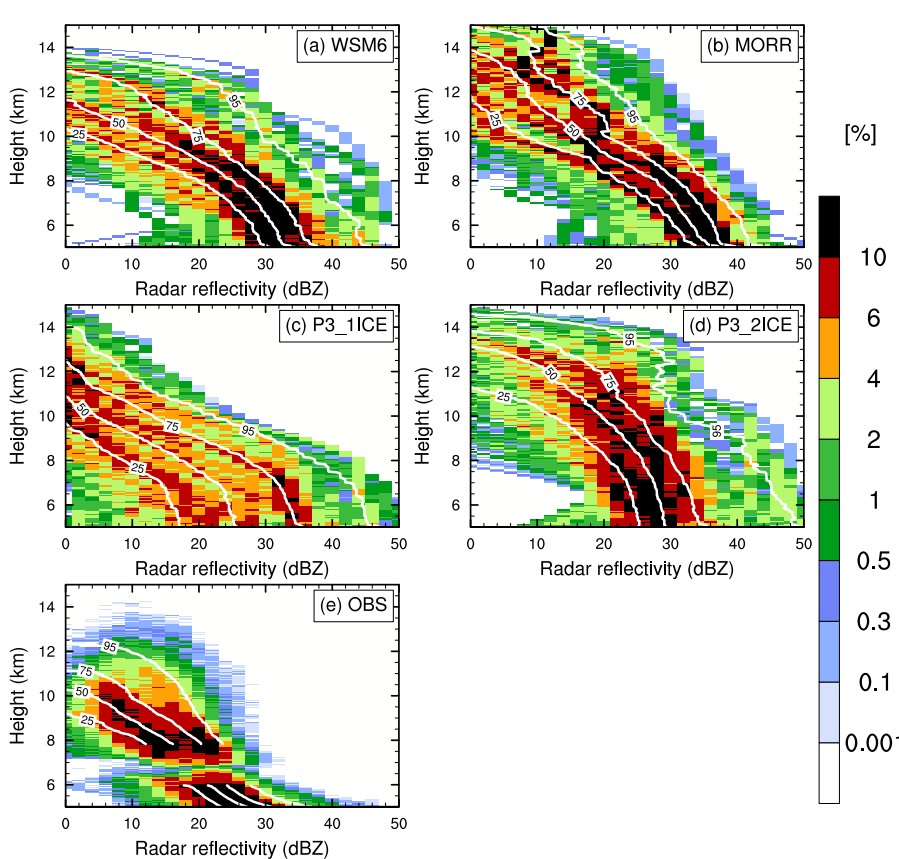

**Figure 7.** Frequency (%, shaded) of simulated (a: WSM6, b: MORR, c: P3-1ICE, d: P3-2ICE) and observed (e) X-band radar reflectivity. Contours represent cumulative frequencies 25%, 50%, 75% and 95%, respectively. The gap at 6–7 km in (e) is due to observed radar reflectivity data not available at the first few range gates (within ∼500 m) from the aircraft.



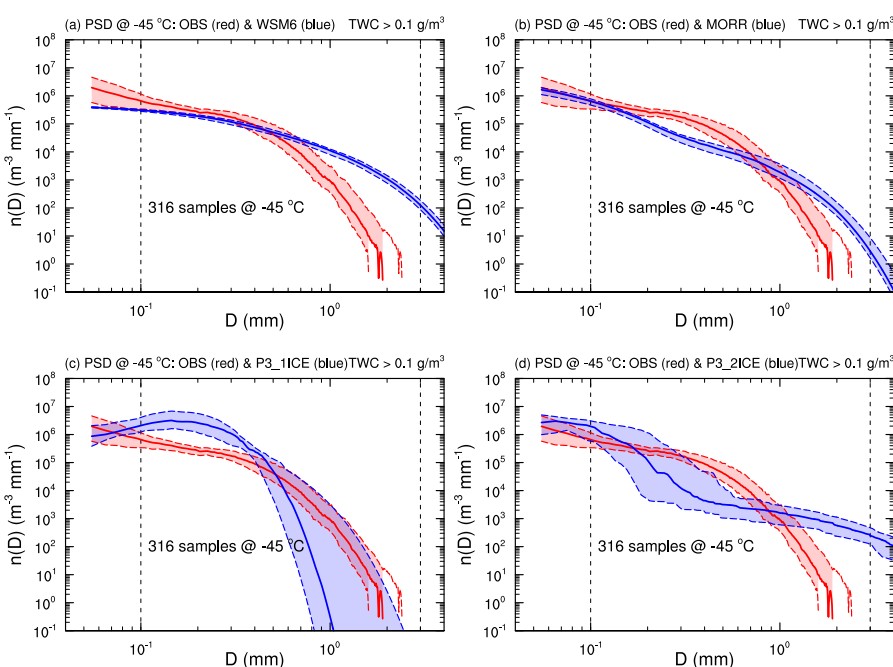

**Figure 8.** Observed (red) and simulated (blue, a: WSM6, b: MORR, c: P3-1ICE, d: P3-2ICE) ice particle size distributions at the level of −45 °C. The red and blue dashed lines indicate the 25th and 75th percentiles, and the red and blue solid lines represent the median.



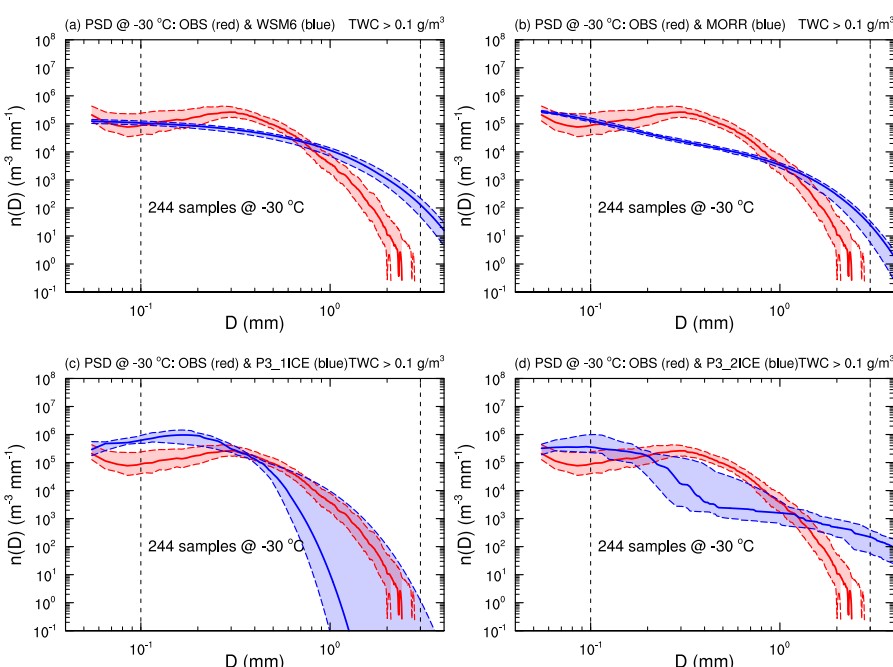

**Figure 9.** Observed (red) and simulated (blue, a: WSM6, b: MORR, c: P3-1ICE, d: P3-2ICE) ice particle size distributions at the level of −30 °C. The red and blue dashed lines indicate the 25th and 75th percentiles, and the red and blue solid lines represent the median.

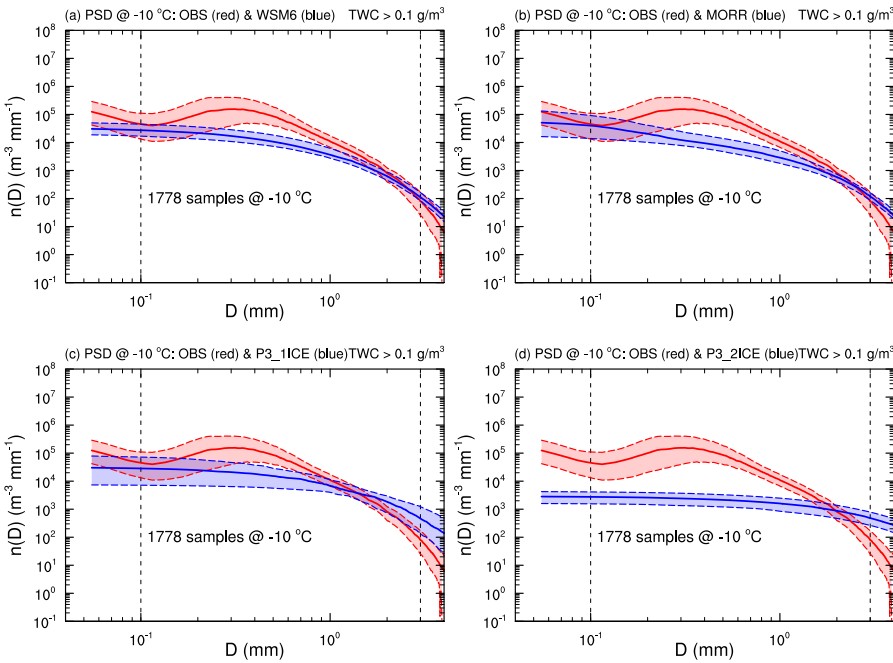

**Figure 10.** Observed (red) and simulated (blue, a: WSM6, b: MORR, c: P3-1ICE, d: P3-2ICE) ice particle size distributions at the level of $-10\ ^{\circ}$C. The red and blue dashed lines indicate the 25th and 75th percentiles, and the red and blue solid lines represent the median.



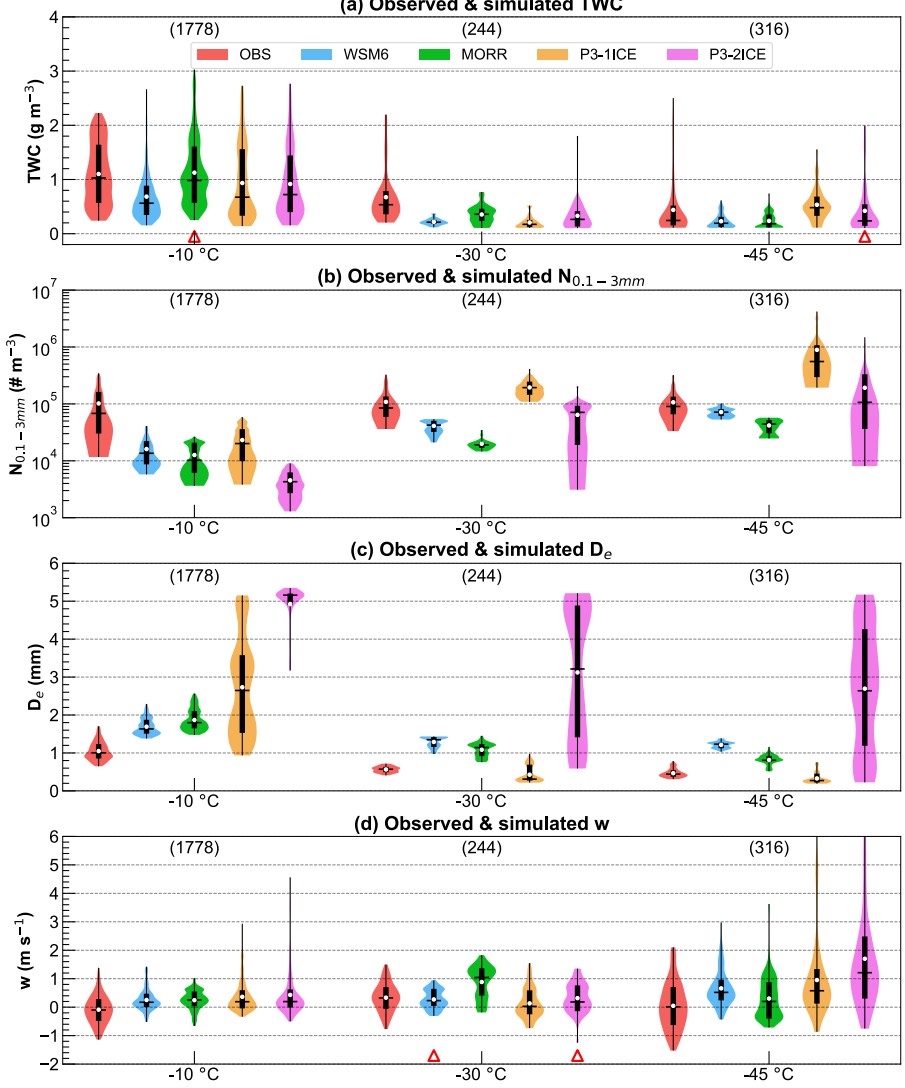

**Figure 11.** Violin plots of observed (red) and simulated (blue: WSM6, green: MORR, orange: P3-1ICE, magenta: P3-2ICE) (a) total water content (TWC, g m$^{-3}$), (b) number concentration (# m$^{-3}$) within 0.1 mm $< D_{max} <$ 3 mm (N$_{0.1-3mm}$), (c) effective diameter ($D_e$, mm), and (d) vertical velocity (m s$^{-1}$) at temperatures of $-10$, $-30$ and $-45$ °C. As for the black box-and-whisker plots, the extremes of the whiskers indicate the 5th and 95th percentiles, the lower and upper limits of the boxes correspond to the 25th and 75th percentiles, the dividing line represents the median value, and the white points represent the average value of samples between the 5th and 95th percentiles. The width of shaded area represents the the proportion of the data located there, and only areas between the 5th and 95th percentiles are shown. The numbers below the upper horizontal axis represent the total number of samples. The red triangles above the bottom horizontal axis indicate that the difference between the experiment mean and the observation mean is not statistically significant (not passing the significant test for p $<$ 0.05).

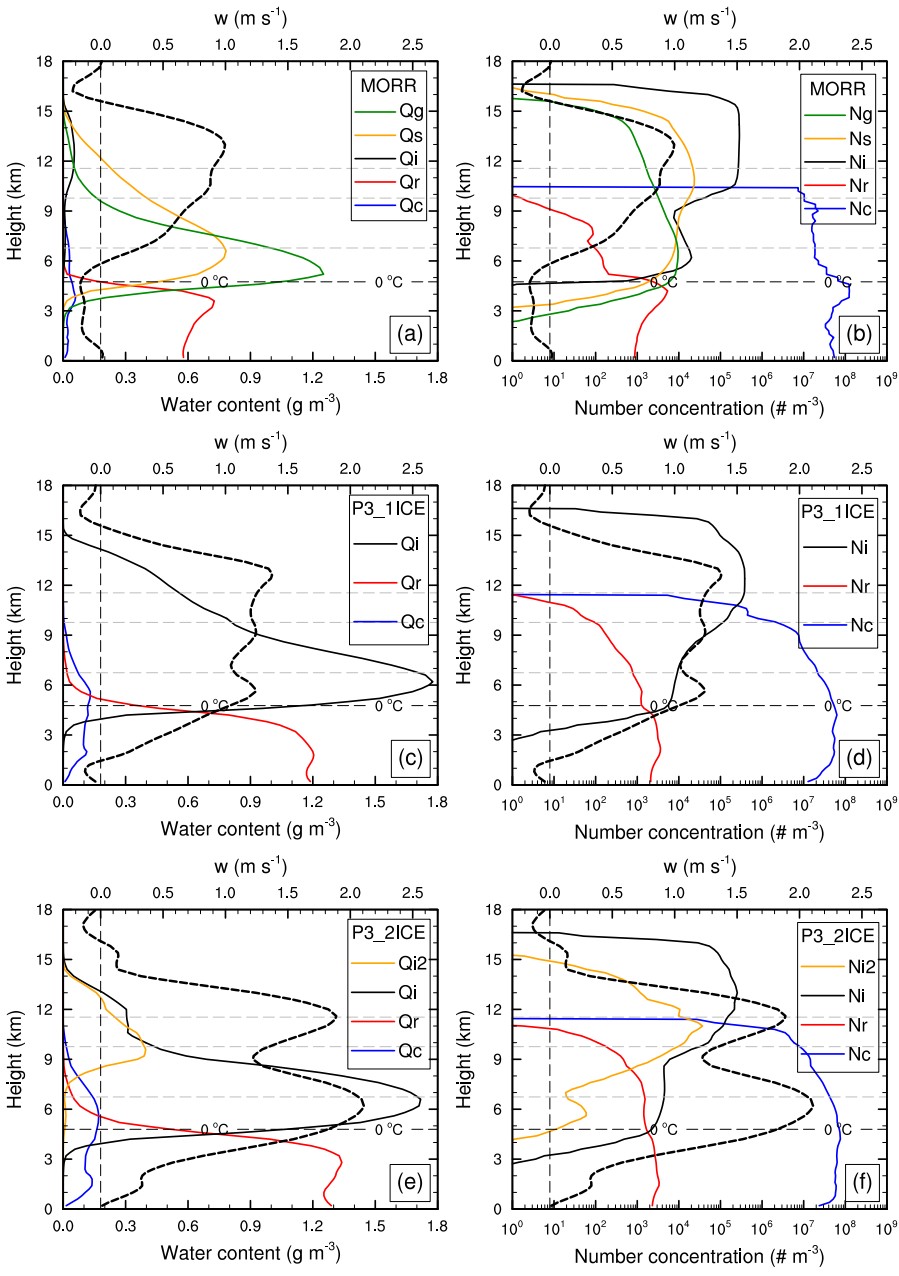

**Figure 12.** Vertical profiles of averaged water content (a, c and e, solid curves, g m$^{-3}$) and total number concentration (b, d, and f, solid curves, # m$^{-3}$) of each hydrometeor category in MORR (a and b, cloud water: blue, rain water: red, cloud ice: black, snow: orange, and graupel: green), P3-1ICE (c and d, cloud water: blue, rain water: red, and cloud ice of first category: black) and P3-2ICE (e and f, cloud water: blue, rain water: red, cloud ice of first category: black, and cloud ice of second category: orange). The thick black dashed lines represent vertical profiles of vertical velocity (m s$^{-1}$). The vertical thin black dashed lines indicate vertical velocity of 0 m s$^{-1}$. The horizontal thin black and gray dashed lines indicate the heights of temperatures at 0, $-10$, $-30$, and $-45$ °C from bottom to top, respectively. The numbers of samples used to calculate the averages are shown in Table 2.



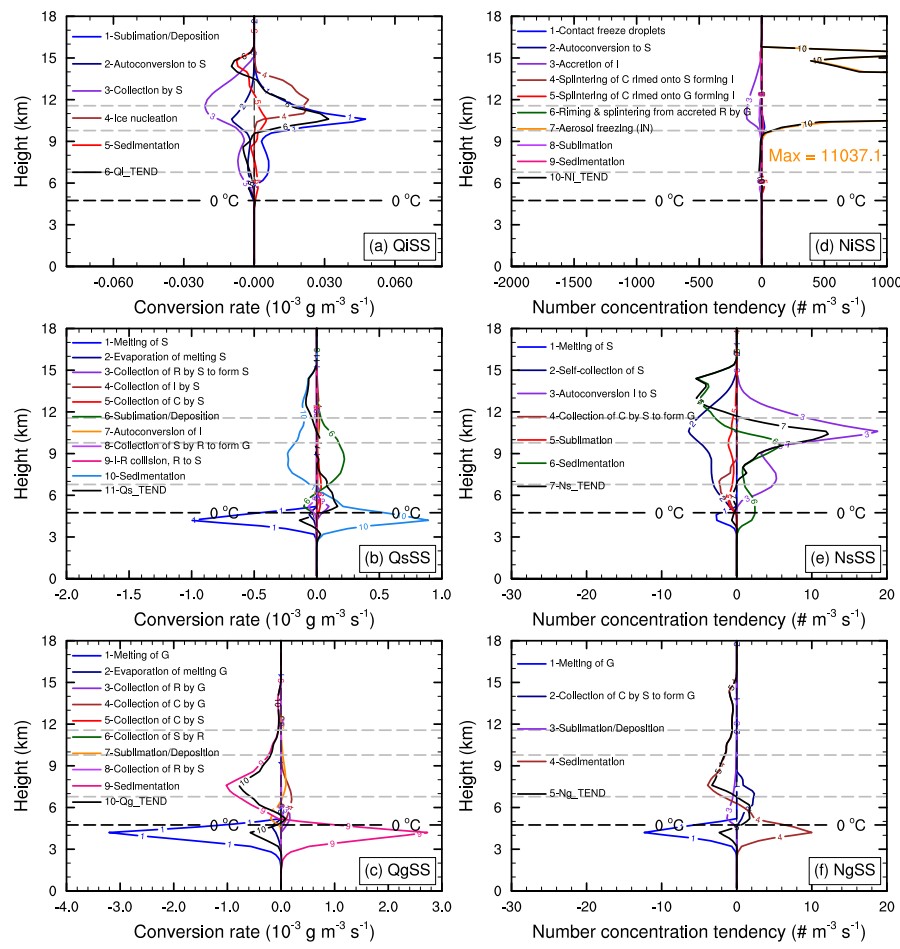

**Figure 13.** Vertical profiles of (a–c) averaged mass conversion rates (solid lines, $10^{-3}$ g m$^{-3}$ s$^{-1}$) and (d–f) averaged number concentration tendencies (solid lines, # m$^{-3}$ s$^{-1}$) of (a, d) cloud ice, (b, e) snow, and (c, f) graupel due to different microphysical processes in MORR. Only the profiles whose column-maximum conversion rates are larger than $10^{-6}$ g m$^{-3}$ s$^{-1}$ and column-maximum number concentration tendencies are larger than 1 m$^{-3}$ s$^{-1}$ are shown in (a–c) and (d–f), respectively. The total microphysics tendencies of cloud ice (a: Qi_TEND, d: Ni_TEND), snow (b: Qs_TEND, e: Ns_TEND) and graupel (c: Qg_TEND, f: Ng_TEND) are shown by black solid curves. The horizontal thin black and gray dashed lines indicate the heights of $0$, $-10$, $-30$, and $-45\ ^{\circ}$C from bottom to top, respectively. The numbers of samples used to calculate the averages are shown in Table 2. C: cloud ice, R: rain water, I: cloud ice, S: snow, and G: graupel.



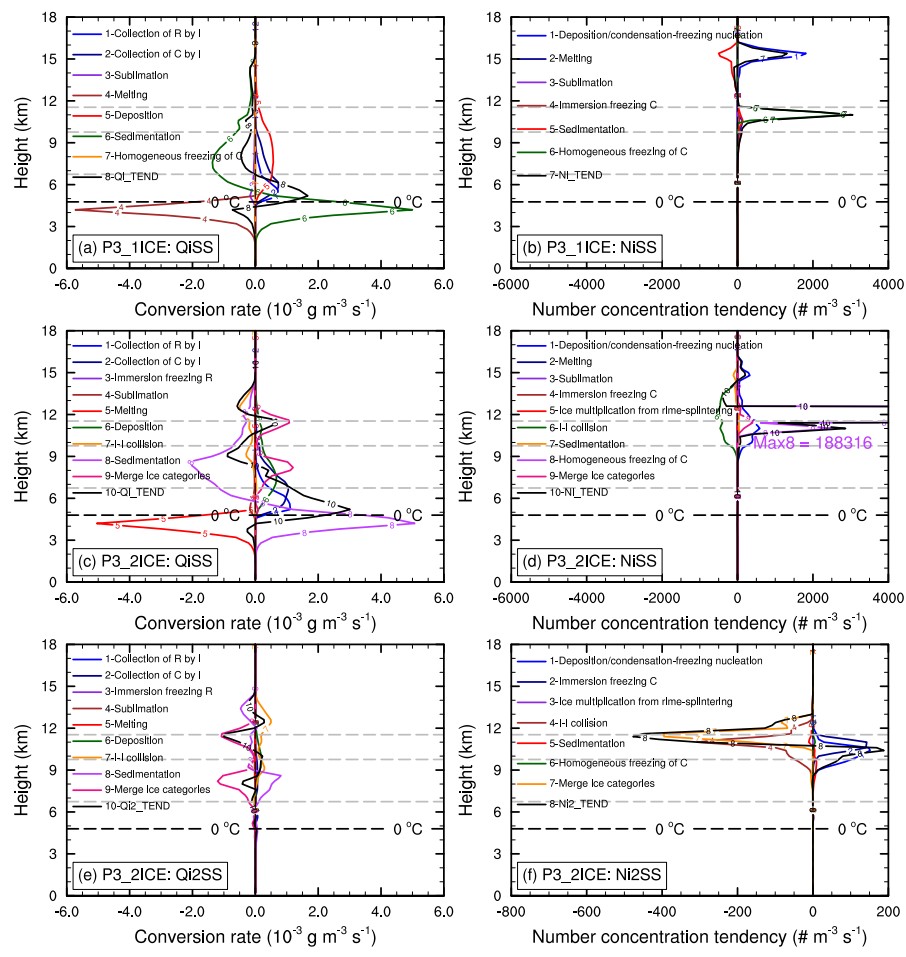

**Figure 14.** Vertical profiles of (a, c and e) averaged mass conversion rates (solid lines, $10^{-3}$ g m$^{-3}$ s$^{-1}$) and (b, d and f) averaged number concentration tendency (solid lines, # m$^{-3}$ s$^{-1}$) of ice categories due to different microphysical processes in (a and b) P3-1ICE and (c–f) P3-2ICE. Only the profiles whose column-maximum conversion rates are larger than $10^{-6}$ g m$^{-3}$ s$^{-1}$ and column-maximum number concentration tendencies are larger than 1 m$^{-3}$ s$^{-1}$ are shown in (a, c and e) and (b, d and f), respectively. The total microphysics tendencies of first ice category (a, c: Qi_TEND, b, d: Ni_TEND), and second ice category (e: Qi2_TEND, f: Ni2_TEND) are shown by black solid curves. The horizontal thin black and gray dashed lines indicate the heights of 0, $-10$, $-30$, and $-45$ °C from bottom to top, respectively. The numbers of samples used to calculate the averages are shown in Table 1. C: ice, R: rain water, I: cloud ice.



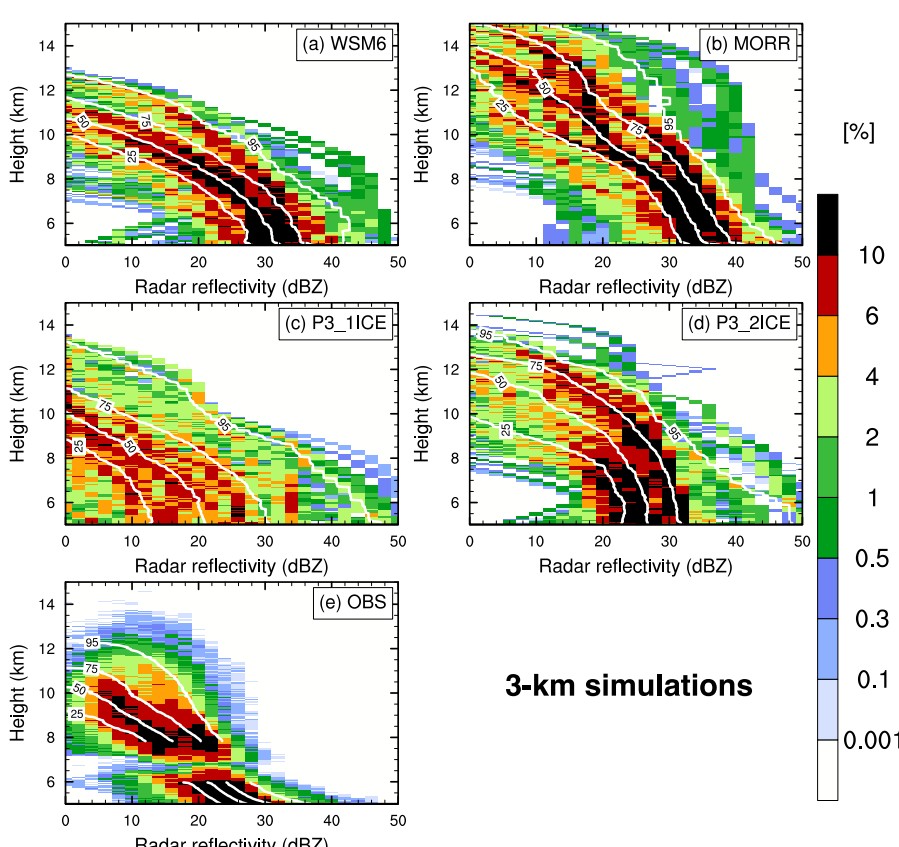

**Figure 15.** As Fig. 7 but for 3-km (d01) simulations.



**Table 1.** Number concentration of median n(D) for 0.1 mm $< D_{max} <$ 12.845 mm ($N_{0.1-12.845mm}$) and contributions of small (0.1 mm $< D_{max} <$ 0.3 mm, $C_{0.1-0.3mm}$), medium (0.3 mm $< D_{max} <$ 1 mm, $C_{0.3-1mm}$), and large (1 mm $< D_{max} <$ 12.845 mm, $C_{1-12.845mm}$) particles at $-45$, $-30$ and $-10$ °C. The bold text indicates the dominant contribution.

| Temp (°C) | Exp | $N_{0.1-12.845mm}$ (# m$^{-3}$) | $C_{0.1-0.3mm}$ (%) | $C_{0.3-1mm}$ (%) | $C_{1-12.845mm}$ (%) |
|---|---|---|---|---|---|
| | Observation | $8.68 \times 10^4$ | **72.0** | 27.9 | 0.1 |
| | WSM6 | $7.18 \times 10^4$ | **54.6** | 38.9 | 6.5 |
| $-45$ | MORR | $4.31 \times 10^4$ | **83.4** | 15.2 | 1.4 |
| | P3-1ICE | $5.32 \times 10^5$ | **86.6** | 13.4 | 0.0 |
| | P3-2ICE | $7.33 \times 10^4$ | **94.7** | 3.0 | 2.3 |
| | | | | | |
| | Observation | $8.59 \times 10^4$ | 42.6 | **56.6** | 0.8 |
| | WSM6 | $4.26 \times 10^4$ | 39.2 | **49.5** | 11.3 |
| $-30$ | MORR | $1.91 \times 10^4$ | **57.0** | 35.8 | 7.2 |
| | P3-1ICE | $1.79 \times 10^5$ | **82.2** | 17.8 | 0.0 |
| | P3-2ICE | $3.66 \times 10^4$ | **89.5** | 6.3 | 4.2 |
| | | | | | |
| | Observation | $6.60 \times 10^4$ | 29.8 | **63.9** | 6.3 |
| | WSM6 | $1.24 \times 10^4$ | 35.4 | **47.9** | 16.7 |
| $-10$ | MORR | $1.05 \times 10^4$ | **42.1** | 40.8 | 17.1 |
| | P3-1ICE | $1.89 \times 10^4$ | 26.0 | **47.5** | 26.5 |
| | P3-2ICE | $0.46 \times 10^4$ | 11.3 | 30.1 | **58.6** |





**Table 2.** Averaged TWC, $N_{0.1-3mm}$ and air vertical velocity (w) at $-10\ ^{\circ}C$. The differences in all averages between the simulations and observations pass the significant test for $p < 0.05$.

| Exp | Sample No. | TWC (g m$^{-3}$) | | $N_{0.1-3mm}$ (# m$^{-3}$) | | w (m s$^{-1}$) | |
|---|---|---|---|---|---|---|---|
| | | OBS[a] | MOD[b] | OBS | MOD | OBS | MOD |
| MORR | 509 | 1.6 | 1.8 | $1.72 \times 10^5$ | $1.78 \times 10^4$ | 0.16 | 0.32 |
| P3-1ICE | 488 | 1.7 | 1.8 | $1.98 \times 10^5$ | $1.25 \times 10^4$ | 0.24 | 1.12 |
| P3-2ICE | 427 | 1.7 | 1.9 | $2.02 \times 10^5$ | $4.63 \times 10^3$ | 0.13 | 2.05 |

[a] Observation, [b] Model.