# Peer review of "Microphysical Processes Producing High Ice Water Contents (HIWCs) in Tropical Convective Clouds during the HAIC-HIWC Field Campaign: Evaluation of Simulations Using Bulk Microphysical Schemes"

_Atmospheric Chemistry and Physics, 2020_

## Referee Comment (RC1) · Anonymous Referee #3 · 8 Dec 2020

The authors propose a study where they compare the ability of 4 different parametrisations of clouds microphysics in WRF, to simulate an extreme weather event during the campaign HAIC-HIWC over Cayenne in May 2015. Attention is focused on the capacity of the 4 schemes to simulate the High Ice Water Content (HIWC) for IWC > 0.1g/m3 with their associated particle size distributions (PSD). Before the presentation of their study, the authors provide a short review of results from former studies on simulation of HIWC and analysis of the HAIC-HIWC dataset. Hence, they highlight

that median mass diameter (MMD) for IWC>1g.m3 increase with the temperature and decrease with increasing total water content (TWC); meaning that HIWC at high altitude are made with small ice crystals. Also, it was showed that X-band radar do not have sensitivity to detect HIWC, but it can be countered using polarimetric parameters such specific differential phase (Kdp) and differential radar reflectivity ratio (Zdr) instead of radar reflectivity factors (Z) only. More importantly, the secondary ice production (SIP) process must have greater contribution in HIWC, than ice nucleating particle to explain their high number concentration of ice particles. A new process of "freezing-drop-shattering" has been recently proposed by Korolev et al. 2020. Simulations with single and double moment microphysical schemes of extreme weather event, explored during HAIC-HIWC in Darwin, showed overestimation of Z in C-band. Also, it showed overestimation of MMD for TWC> 1g/m3. Another simulation of an event during HAIC-HIWC in Cayenne, failed to represent observed IWC and PSD, suggesting a lack of the representation of SIP processes. In the presented study, there is no new parametrisation of clouds microphysics tested. But, two parametrisations are tested for the first time to simulate HIWC, where the density of the ice is predicted for one or two ice species (called in the study P3-1ICE and P3-2ICE). The other microphysical scheme are a single moment with 6 species of hydrometeors (WSM6) and a double moment with 5 hydrometeors types (MORR). Quality of simulated HIWC and PSD is tested for each scheme, through the comparison with observed Brightness temperature (Tb) of the channel 4 of GOES-13, the observed radar reflectivity factors (Z) in the X-band and the observed PSD. The authors conclude a good prediction in average of temperature, dew point and winds fields, where prediction with MORR are closer to observations. The comparison of observed brightness temperature with the simulations shows better results with the scheme P3-2ICE. However, simulated Z with the four microphysical scheme are larger than the Z observed. The four microphysical schemes give bad representation of particle size distributions when compared with the observations. Usually, total concentration of ice crystals are underestimated except for P3-1ICE at -30°C and -45°C. In the prediction, mixed-phased at -10°C processes lead to an overestimation

of LWC.

Major Comments: In this study, it is difficult to see the link between HIWC and their associated microphysical processes. Main of the figures and conclusion are built on figures that includes IWC>0.1g/m3, while the authors give the definition of HIWC such IWC>1g/m3. The comparison of size distribution is made for average PSD and for IWC > 0.1g/m3, which is a large spread. This study should at least separate the results with on one side figures and comment for 0.1g/m31g/m3. In the introduction, the authors bring the attention on what HIWC at high altitudes are made with small ice crystals ($\sim$300-400$\mu$m) and that X band are not suited for detection of HIWC. But in the present study, the authors still use the X-band radar to evaluate the distribution of IWC in their simulations, while a look in the former publication that dedicated studies on HAIC-HIWC dataset cited in the introduction, give the information that a cloud radar was on board with the microphysical probes. Knowing the size of ice crystals that made mainly HIWC and the strong relationship demonstrated between Z, IWC and T, cloud radar is more suited to evaluate the distribution of IWC. Comparison of Z in X-band only allows to study aggregates densities and their distribution, thus aggregation process and precipitation. As Z at these frequencies ($\sim$10GHz) are more sensitive to the concentrations of large hydrometeors and not the total water content itself (see Drigeard et al., 2015). Overall, the conclusions are mainly known, overestimation of Z in C/X-bands, overestimation of LWC and then too much riming (see cited publications in the introduction; C and X bands differs from their radar constant and attenuation, but their response with regards to the hydrometeors are similar) and mainly underestimation of total concentrations due to a lack of SIP processes in convective clouds. However, the fact that P3-1ICE and P3-2ICE can produce high concentration of ice crystals is interesting. What SIP these two schemes take into account? The authors mentioned in the introduction a companion paper more dedicated to the schemes P3. I suggest that the actual paper is withdrawn and will be re-submitted in the same time than its companion paper, including in their titles "part 1" and "part 2".

Minor comments: Page 2, line 43-49: Does this IWC-Z-T relationship is in X or C band? How does it compares with the one of Protat et al., 2016 cited in line 35 of the same page? Same for the methodology from Nguyen et al. 2019, how does it compared with the one of Protat et al. 2016, mean bias, rms of both methods? Moreover, authors do not use X-band polarized parameters as it seems to be suggested by Nguyen et al. 2019 in order to study IWC, why? Page 5-6, Section 3.2: It would help to add a description of main microphysical process and the SIP that are taken into account by each schemes. It will helps later, to discuss why P3 schemes have such high concentrations at high altitudes (Figure 11b)? Page 6, section 3.3: As commented in the major comment, evaluation of only Z in X-band can only help to study aggregation and precipitation processes. W-band is more suited for the topic of this study. Page 7, lines 187-215: Is there a relation between cloud top temperature and the IR brightness temperature? Does it means that the 4 schemes have a good predictions of the height of the convective clouds? What are cloud top heights that the radar estimate with regards to the co-located Tb? Page 8, lines 216-253: Does the same range of IWC in observations and predictions give the same range of Z, for the same T. It may need to be completed with cloud radar observations and comparison? Page 9, section 4.2: from figure 11a, it seems that range of IWC are not exactly the same with similar distributions as function of microphysical schemes and observations. The radar comparison shows a large variability of Z and a distribution of Z different between observation and prediction. Does it make sense to compare PSD over all IWC? I suggest a comparison of PSD as function of IWC range? The overestimation of concentrations of large hydrometeors can explain the figure 7. The four scheme could predict similar density as in observation, but the fact that the prediction of concentration of large hydrometeors are larger than in the observations is enough to understand the overestimation of Z ? What is the impact of IWC in the overestimation of Z? Note that P3-1ICE at -30°C and -45°C predict similar concentrations, or more exactly share a similar range of concentration with observations for large hydrometeors and that predicted Z are closer with the one observed (figure 7) than the other predicted Z. Page 14, line 417: Is this a new

results?

Page 16, equation A2 + Figure 11b: Does the total concentration presented in Figure 11 from WRF prediction are calculated with this equation too ? Th comparison would not be fair if it is.

---

## Referee Comment (RC2) · Anonymous Referee #1 · 7 Jan 2021

The paper reports on a regional simulation of an MCS where large ice water contents were observed. Four different microphysical schemes were compared to observations. The paper is clear and to the point. I think that only minor revisions are necessary before publishing.

This is obviously a nice test case and one that others may try to reproduce themselves. The observations are freely available, but will scripts to reproduce the sampling outlined

here also be available?

Is there an accepted definition for HIWC region. Can this be indicated and compared in some way to show that models have some skill at predicting these regions? Including something like this would link the results nicely to the operational motivation for this work.

line 55. Could refer to Keinert et al. who have carried out laboratory experiments to investigate droplet freezing/shattering in this temperature range. (https://www.researchgate.net/publication/343566907_Secondary_Ice_Production_upon_Freezing_of_Freely_Falling_Driz

line 159. Feel free to ignore this because there are always more that can be added to an intercomparison, but given the operational importance of HIWC events i'm surprised that the Thompson scheme was also not included in the mix - i believe it is operationally used (or was) in the NOAA RUC model.

line 182 - '...likely associated with..' - i think you should be able to say yes or no to this by inspecting era data rather than leaving it hanging.

line 189 'using the assumptions consistent' - does this include the shape of the psd or just the mass-size/density assumptions?

line 207-210. Should be careful to add that your statement is for this metric: BT. e.g. this means deep convective areas as defined by this BT metric are larger in MORR than.... You could define it by updraft or specific humidity...

line 230. it may be too messy but it might be worth trying to add the cumulative frequency contours from the obs to the model panels to provide an easier way to compare across?

line 250. even though there is a bias in sampling - is this not compensated with the BT sampling methodology?

line 252. can you estimate the impact? If not i think you have to assume its unbiased...?

line 284. how is psd spread quantified?

line 302. reasonably to within x% ?

line 311. Possibly add a radar weighted mean size to link to radar results?

line 333. 'it is found that the main microphysical process rates at -45 and -30C are the same a those within profiles containing HIWC regions at -10C'. I don't think you mean this but i'm interpreting it as using process rates at -10C as a proxy for what is going on at -45 and -30C. If so, then i would expect processes involving graupel production to be different between -10C and -45C.

Additionally, in strong convection, the loss of liquid at lower levels controls the liquid being transported to higher up and the eventual anvil evolution.

At -45C the freezing is dominated by homogeneous freezing, whereas at -10C it will be heterogeneous freezing or secondary ice production. Therefore i don't think you can use the process rates at -10C as proxy for -45C.

line 335. i think i disagree here. Transporting more cloud droplets to homogeneous freezing altitudes/temperatures could lead to more numerous small ice crystals.

line 349-350 '...substantially underpredict the ice particle number for 0.1 mm < Dmax < 3 mm and overpredict the vertical motion in the HIWC regions, which results in stronger and higher-extended simulated radar reflectivity...' and line 355-356 '...an underestimate of ice particle number concentration, especially graupel, leads to large reflectivities...'

Because the psd is a gamma or exponential distribution in the model i accept that reducing total number concentration will lead to an increase in the reflectivity for a fixed mass. But invoking that strong correlation between number concentration dominated by the small end of the PSD and the large end of the PSD that affects the radar reflectivity is not a given for the real world.

[Figure]

In the real world, underpredicting number concentration will not necessarily be the direct source of the radar reflectivity overestimate - its the SUM_i(m_i^2) integral where i is the i_th size bin. If the total mass is correct the overestimate of Ze must come from the the mass being in too large size bins as your PSD comparisons suggest.

I can see that in the appendix the radar reflectivity is formulated to depend upon Nt due to the gamma assumption, but it feels more physical to relate the effect to the large end of the PSD.

line 354 - do you need to add a total ice category line to the MORR plot to compare to P3?

line 371. you could resample the data to match the liquid water content from the model and observations and then see if the ice properties etc are biased.

line 375. ice nucleation = homogeneous freezing? or hom+het freezing?

line 378. i could not really see this figure - all i can see is the total ni_tend going out of range. It looks like a rime splintering secondary ice production is represented but has no effect?

line 451. the 3km radar results should appear earlier than the final page i think.

line 459 it doesnt look that clear cut to me. based on red triangles in fig 11 i score it as p3-2ice=2, wsm6=1, morr=1

The psds using the mean of the De metric: -10C wsm6, morr, p3_1, p3_2 -30C p3_1, morr, wsm6, p3_2 -45C p3_1, morr, wsm6, p3_2

cfads - the cumulative curves from p3_1 seem to match best with obs. then wsm6, morr, p3_2

If you are going to say which is best i think you need some quanitative measures to quote.

---

## Author Comment (AC1) · 17 Feb 2021

**Responses to the comments of Referee #1**

Referee #1: The paper reports on a regional simulation of an MCS where large ice water contents were observed. Four different microphysical schemes were compared to observations. The paper is clear and to the point. I think that only minor revisions are necessary before publishing.

**Response:** We would like to express our acknowledgement for your efforts and constructive comments. Our point-by-point responses are given below. For convenience, the reviewers' comments are in **black** fonts, and our point-by-point responses are in **blue**.

This is obviously a nice test case and one that others may try to reproduce themselves. The observations are freely available, but will scripts to reproduce the sampling outlined here also be available?

**Response:** The scripts used in our study do not have detailed comments currently, so someone who is interested in this study and wants to reproduce the sampling can contact us directly, and we will give some instructions how to use the scripts.

Is there an accepted definition for HIWC region. Can this be indicated and compared in some way to show that models have some skill at predicting these regions? Including something like this would link the results nicely to the operational motivation for this work.

**Response:** The broad definition of a HIWC region is "Regions with high ice water content (HIWC), composed of mainly small ice crystals, frequently occur over convective clouds in the tropics. Such regions can have median mass diameters (MMDs) < 300 µm and equivalent radar reflectivities < 20 dBZ". However, currently there is no unified definition for the threshold of IWC, and the threshold of 1 g cm-3 is used in our study. From the simulations in this study, regions with higher IWC can be predicted by the model, while reflectivities of these regions are overestimated. It indicates that model can capture the regions with higher IWC, but it cannot represent the phenomenon of numerous small ice crystals contributing to the IWC. This is summarized in the last section.

line 55. Could refer to Keinert et al. who have carried out laboratory experiments to investigate droplet freezing/shattering in this temperature range.

(https://www.researchgate.net/publication/343566907\_Secondary\_Ice\_Production\_upon\_Fre ezing\_of\_Freely\_Falling\_Drizzle\_Droplets)

**Response:** We have cited the paper in our revised manuscript.

line 159. Feel free to ignore this because there are always more that can be added to an intercomparison, but given the operational importance of HIWC events I'm surprised that the Thompson scheme was also not included in the mix - I believe it is operationally used (or was) in the NOAA RUC model.

**Response:** We selected microphysics schemes for study that parameterized ice species differently. WSM6 is a single-moment scheme for cloud ice, snow and graupel; the Morrison scheme is a double-moment scheme for cloud ice, snow, and graupel; P3 represents ice species very differently. The Thompson scheme uses a double moment for cloud ice and a single moment for snow and graupel. If possible, we will add the Thompson scheme in our future modeling studies.

line 182 - '...likely associated with..' - i think you should be able to say yes or no to this by inspecting era data rather than leaving it hanging.

Response: We have revised the statement and removed the word "likely".

line 189 'using the assumptions consistent' - does this include the shape of the psd or just the mass-size/density assumptions?

**Response:** Yes, it includes the parameters of the PSDs in addition to mass-size/density assumptions. We have added a description "including the characteristics of the cloud species and PSDs" in our revised manuscript to clarify this point.

line 207-210. Should be careful to add that your statement is for this metric: BT. e.g. this means deep convective areas as defined by this BT metric are larger in MORR than.... You could define it by updraft or specific humidity...

**Response:** We have revised the statement to make it clear that it is for the BT metric, which now reads "This means deep convective areas as defined by BT metric are larger in MORR than the observations at an early stage in the system."

line 230. it may be too messy but it might be worth trying to add the cumulative frequency contours from the obs to the model panels to provide an easier way to compare across?

**Response:** The white contours in Fig. 7 represent cumulative frequency. We have revised the description to make this more clear.

line 250. even though there is a bias in sampling - is this not compensated with the BT sampling methodology?

**Response:** Yes, the BT sampling method can compensate for part of the sampling bias. We have revised the related sentences in our manuscript, which now reads "It should be noted that the NRC Convair 580 operations avoided the cloud regions with high reflectivity due to safety regulations, and thus it did not approach high reflectivity regions (red zones on the pilot's radar) within 30 nautical miles (~55.56 km). However, the BT sampling method has been used to minimize these aircraft sampling biases."

line 252. can you estimate the impact? If not i think you have to assume its unbiased...?

**Response:** We cannot estimate the impact. But, we have revised the manuscript to emphasize that the BT sampling method minimizes the sampling biases, which now reads "However, the BT sampling method has been used to minimize these aircraft sampling biases."

line 284. how is psd spread quantified?

**Response:** The "spread" here means the range between the 25th and 75th percentiles in Fig. 10b. We have indicated this in our revised manuscript, which now reads "…same PSD spread (range between the 25th and 75th percentiles)…".

**line 302. reasonably to within x%?**

**Response:** We have updated the related statement, which now reads "Generally all the simulations, especially MORR, reproduce the TWC reasonably within the same order of magnitude as observations at the three temperature levels, with biases within 38% at  $-10 \circ$ C"

line 311. Possibly add a radar weighted mean size to link to radar results?

**Response:** We should clarify that observed radar reflectivity profiles were sampled along the flight track of NRC Convair 580, whose horizontal locations are different from those of observed TWC/PSD samples from SAFIRE Falcon 20 (Fig. 1). Part of observed TWC/PSD samples at -10 °C and all observed TWC/PSD samples at -45 and -30 °C are from SAFIRE Falcon 20 (Fig. 1), implying that observed TWC/PSD at -45 and -30 °C may be not fully consistent with observed radar reflectivity in the upper levels in this study. Based on this bias, we cannot link the radar weighted mean size to radar results directly.

line 333. 'it is found that the main microphysical process rates at -45 and -30C are the same as those within profiles containing HIWC regions at -10C'. I don't think you mean this but i'm interpreting it as using process rates at -10C as a proxy for what is going on at -45 and -30C. If so, then i would expect processes involving graupel production to be different between -10C and -45C. Additionally, in strong convection, the loss of liquid at lower levels controls the liquid being transported to higher up and the eventual anvil evolution. At -45C the freezing is dominated by homogeneous freezing, whereas at -10C it will be heterogeneous freezing or secondary ice production. Therefore i don't think you can use the process rates at -10C as proxy for -45C.

**Response:** Our original description confused both reviewers so we have reworded this. We meant that the main microphysical processes are similar at the same vertical levels when comparing sampling profiles with TWC > 1 g m-3 at -10 °C to sampling profiles with TWC > 0.1 g m-3 at -45 and -30 °C. We have revised the manuscript to make this more clear, which now reads "Through comparing between simulated sampling profiles with TWC > 0.1 g m-3 at -45 and -30 °C. We have revised the manuscript to make this more clear, which now reads "Through comparing between simulated sampling profiles with TWC > 0.1 g m-3 at -45 and -30 °C. We have revised the manuscript to make this more clear, which now reads "Through comparing between simulated sampling profiles with TWC > 0.1 g m-3 at -45 and -30 °C. Two sampling profiles with TWC > 1 g m-3 at -10 °C, it is found that their main microphysical processes are the same at the same vertical levels."

line 335. i think i disagree here. Transporting more cloud droplets to homogeneous freezing altitudes/temperatures could lead to more numerous small ice crystals.

**Response:** We have deleted this sentence as we agree it is not fully validated by the available data.

line 349-350 '...substantially underpredict the ice particle number for 0.1 mm